# Risk of the hydrogen economy for atmospheric methane

**Matteo B. Bertagni** [1] ✉, **Stephen W. Pacala**[2], **Fabien Paulot** [3] &
**Amilcare Porporato** [1,4]

Hydrogen ($H_2$) is expected to play a crucial role in reducing greenhouse gas emissions. However, hydrogen losses to the atmosphere impact atmospheric chemistry, including positive feedback on methane ($CH_4$), the second most important greenhouse gas. Here we investigate through a minimalist model the response of atmospheric methane to fossil fuel displacement by hydrogen. We find that $CH_4$ concentration may increase or decrease depending on the amount of hydrogen lost to the atmosphere and the methane emissions associated with hydrogen production. Green $H_2$ can mitigate atmospheric methane if hydrogen losses throughout the value chain are below $9 \pm 3\%$. Blue $H_2$ can reduce methane emissions only if methane losses are below 1%. We address and discuss the main uncertainties in our results and the implications for the decarbonization of the energy sector.

Commitments to reach net-zero carbon emissions have drawn renewed attention to hydrogen ($H_2$) as a low-carbon energy carrier[1,2]. Currently, $H_2$ is mostly used as an industrial feedstock, and its global production has a high carbon footprint because it relies almost entirely ($\approx$95%) on fossil fuels[1]. However, many technologies to produce $H_2$ with a lower carbon footprint are available[1]. Among these, low-carbon $H_2$ can be produced from water electrolysis powered by renewable energy (green $H_2$) or from methane reforming coupled with carbon capture and storage (blue $H_2$). $H_2$ fuel may be especially important to decarbonize energy and transport sectors where direct electrification is complicated, like heavy industry, heavy-duty road transport, shipping, and aviation[1]. $H_2$ is also being considered for storing renewable energy[1]. As a result of this potential, countries accounting for more than a third of the world's population have developed national strategies for large-scale $H_2$ production[1,2].

Even if a more hydrogen-based economy would reduce $CO_2$ emissions and improve air quality[3], it would also increase the $H_2$ emissions into the atmosphere. The $H_2$ molecule is very small and difficult to contain, so it is still largely unknown how much $H_2$ will leak in future value chains. $H_2$ emissions will also occur due to venting, purging, and incomplete combustion[4–6]. This potential increase in $H_2$ emissions has received relatively little attention to date because $H_2$ is neither a pollutant nor a greenhouse gas (GHG). However, it has been long known[7–10] that $H_2$ emissions may exert a significant indirect radiative forcing by perturbing the concentration of other GHG gases in the atmosphere. This indirect GHG effect of $H_2$ calls for a detailed scrutiny of the global $H_2$ budget and the environmental consequences of its perturbation[11,12].

$H_2$ is the second most abundant reactive trace gas in the atmosphere, after methane, with an average concentration of around 530 $ppb_v$[13]. $H_2$ sources include both direct emissions ($\approx$45% of total sources) and production in the troposphere from the oxidation of volatile organic compounds ($\approx$25%) and methane ($\approx$30%)[11,14]. The main $H_2$ sinks are the uptake by soil bacteria (70–80% of total tropospheric removal) and the atmospheric reaction with the radical OH (20–30%), which is responsible for the indirect GHG effect of $H_2$. $H_2$'s reaction with the OH radical tends to increase tropospheric methane ($CH_4$) and ozone ($O_3$), which are two potent greenhouse gases. It also increases stratospheric water vapor, which is associated with stratospheric cooling and tropospheric warming[8,15]. Recent global climate models have estimated that hydrogen has an indirect radiative forcing of around $1.3$[14]–$1.8$[16] $10^{-4}$ W m$^{-2}$ $ppb_v^{-1}$, and a global warming potential (GWP) that lies in the

[1]The High Meadows Environmental Institute, Princeton University, Guyot Hall, Princeton 08544 NJ, USA. [2]Department of Ecology and Evolutionary Biology, Princeton University, Guyot Hall, Princeton 08544 NJ, USA. [3]Geophysical Fluid Dynamics Laboratory, National Oceanic and Atmospheric Administration, 201 Forrestal Rd, Princeton 08540 NJ, USA. [4]Department of Civil and Environmental Engineering, Princeton University, Guyot Hall, Princeton 08544 NJ, USA. ✉e-mail: matteobb@princeton.edu

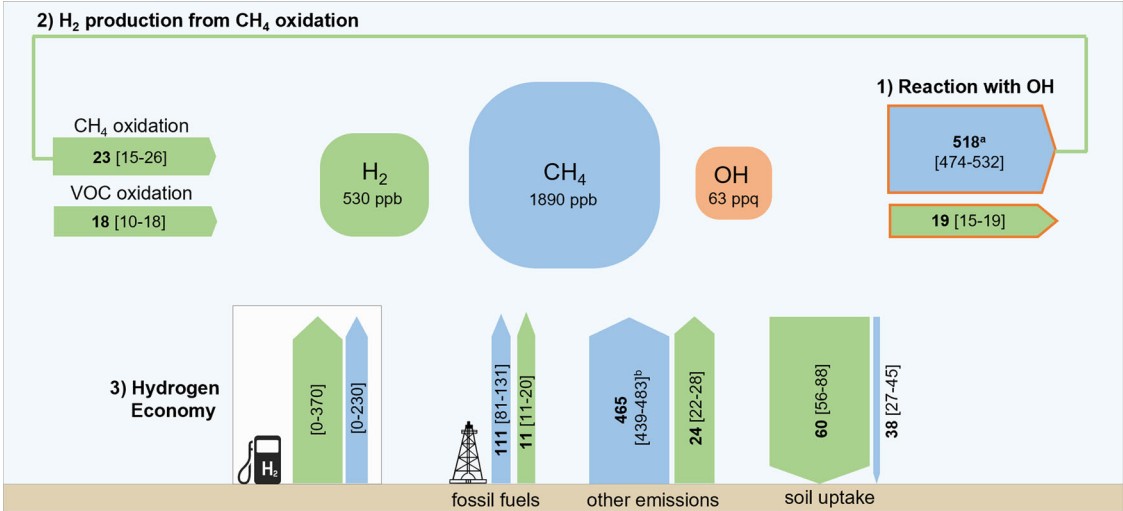

**Fig. 1 | Tangled hydrogen (H₂) and methane (CH₄) budgets.** Sketch of $H_2$ and $CH_4$ tropospheric budgets and their interconnections: (1) the competition for OH; (2) the production of $H_2$ from $CH_4$ oxidation; (3) the potential emissions [minimum-maximum] due to a more hydrogen-based energy system. Flux estimates (Tg/year) are from refs. 11,18. Arrows are scaled with mass flux intensity, $CH_4$ scale being 10 times narrower than $H_2$ scale. On a per-mole basis, $H_2$ consumes only around 3 times less OH than $CH_4$. ppq = part per quadrillon ($10^{-15}$). $^a$ top-down estimate including also minor atmospheric sinks (<10%). $^b$ range obtained as a difference between total and fossil fuel emissions[18].

range 11 ± 5 for a 100-year time horizon[16]. Hence, $H_2$ emissions are far from being climate neutral, and their largest impact is related to the perturbation of atmospheric $CH_4$[14,16], the second most important anthropogenic GHG.

The tropospheric budgets of $H_2$ and $CH_4$ are deeply inter-connected (Fig. 1). First, the removal of both gases from the atmosphere is controlled by their reaction with OH, which is the dominant sink (≈90%) for atmospheric methane[17,18]. An increase in the concentration of tropospheric $H_2$ may reduce the availability of OH, consequently weakening $CH_4$'s removal and increasing $CH_4$'s lifetime and abundance[14,19]. Second, methane is a primary precursor of hydrogen. Namely, $CH_4$ oxidation results in the production of formaldehyde, whose photolysis produces $H_2$. Firn-air records suggest that the increase in $H_2$ over the 20th century can be largely explained by the increase in $CH_4$ concentration[20].

Additionally, $H_2$ and $CH_4$ are linked at the industrial level. Around 60% of global $H_2$ production is currently produced from steam methane reforming (gray $H_2$) and is responsible for 6% of global natural gas use[1]. In the next decade, steam methane reforming coupled with carbon capture and storage will likely remain the dominant technology for large-scale $H_2$ production (blue $H_2$), since facilities for $H_2$ production from renewable sources (green $H_2$) will require time to become operational and economically favorable[2].

Since $CH_4$ is the second-largest contributor to atmospheric warming since the beginning of the industrial era and there are global efforts to mitigate its atmospheric levels[21], it is crucial to quantify the response of atmospheric $CH_4$ to increasing $H_2$ production.

We analyze this problem through a simple atmospheric model that captures the interaction between $H_2$ and $CH_4$ ("Methods"). The investigation of the transient dynamics ("Methods") shows that any $H_2$ emissions pulse to the atmosphere leads to a small transient growth of atmospheric $CH_4$ whose effects last for several decades. In the next sections, we focus on how the equilibrium concentrations of tropospheric $H_2$ and $CH_4$ would respond to scenarios of continuous emissions from an energy system where part of the fossil fuel energy share is replaced by green or blue $H_2$. The analysis emphasizes how atmospheric $CH_4$ could either decrease or increase, mainly depending on the $H_2$ production pathway and the amount of $H_2$ lost to the atmosphere. The latter is defined through the hydrogen emission intensity (HEI), namely the percentage of $H_2$ produced that is lost to the atmosphere. Specifically, we find a critical HEI above which the $CH_4$ atmospheric burden rises despite the lower fossil fuel use. We assess the critical factors and the main uncertainties in the quantification of this critical HEI. We finally discuss how our results can help better inform policymakers regarding the trade-off associated with different scenarios of hydrogen production and use.

## Results
### Emission scenarios

Here we investigate how the tropospheric burdens of methane and hydrogen would be affected by the transition to a more hydrogen-based energy system, wherein hydrogen replaces part of the current fossil fuel energy (≈490 ExJ in 2019[22]). To achieve this goal, we estimate the $CH_4$ and $H_2$ source changes, $\Delta S_{CH_4}$ and $\Delta S_{H_2}$, where $\Delta$ indicates the difference to the current tropospheric conditions ("Methods"). This fossil fuel displacement reduces both $CH_4$ and $H_2$ sources (Fig. 1). The rise in $H_2$ production causes additional $H_2$ emissions due to intentional (e.g., venting) and unintended (e.g., fugitive) losses, and possibly $CH_4$ emissions associated with blue $H_2$ production.

The change in $H_2$ emissions can be estimated from the amount of hydrogen produced to substitute fossil fuels and the HEI, namely the percentage of $H_2$ produced that is lost to the atmosphere. Losses can occur due to venting, purging, incomplete combustion and leaks across the hydrogen value chain. The HEI of the future global $H_2$ value chain is very uncertain. Literature values range from 1 to 12%[4,9,23], but the upper bound is unlikely to occur at large scales because it would be both unsafe and too expensive. Recent empirical estimates for specific $H_2$ infrastructures suggest HEI's ranging from 0.1 to 6.9%, critically depending on the pathway of hydrogen production and transport[6]. To account for these uncertainties and to explore a broad spectrum of possible scenarios, here we vary HEI from 0 to 10% of the total hydrogen produced (Fig. 2a). The lower and upper bounds of this range represent a perfectly sealed and a highly leaking global $H_2$ value chain, respectively. With a perfectly sealed hydrogen value chain, $H_2$ emissions would only decrease due to the lower fossil fuel use. On the contrary, a highly leaking $H_2$ value chain, coupled with an envisioned penetration of $H_2$ in the energy market, could increase hydrogen emissions up to several times the total current sources, which are around 80 Tg $H_2$ yr$^{-1}$.

The variation in $CH_4$ emissions depends not only on the percentage of fossil fuel energy that is displaced by hydrogen, but also on the

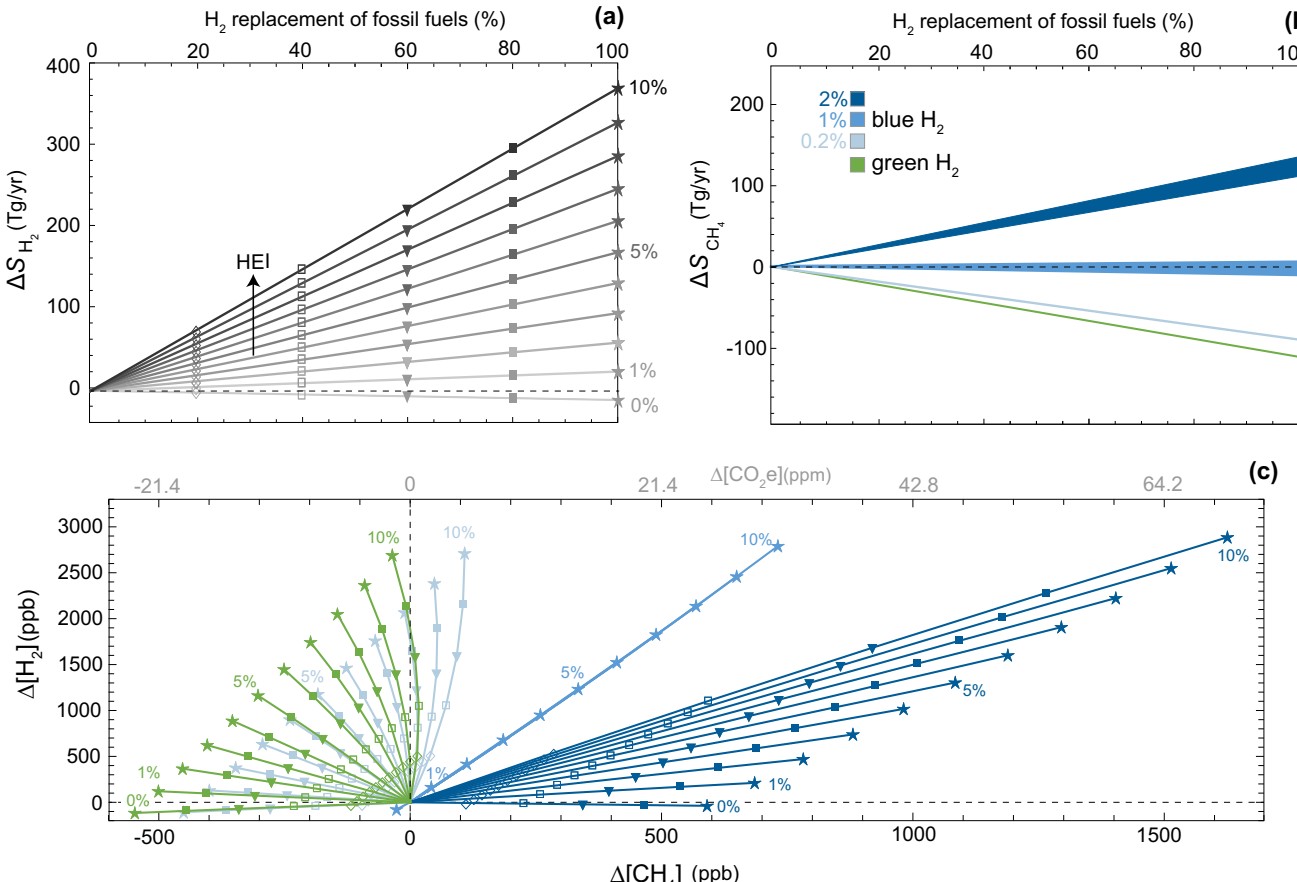

**Fig. 2 | Hydrogen replacement of fossil fuels. a** Changes in $H_2$ sources ($\Delta S_{H_2}$) as a function of fossil fuel replacement for different hydrogen emission intensity (HEI). **b** Changes in $CH_4$ sources ($\Delta S_{CH_4}$) as a function of fossil fuel replacement for different $H_2$ production pathways. Methane leak rates associated with blue $H_2$ production are 0.2, 1, and 2%. Bands for $\Delta S_{CH_4}$ account for different amounts of blue $H_2$ produced and lost. **c** Response of the tropospheric concentrations of $H_2$ and $CH_4$ for the emission scenarios of the previous panels. Symbols mark the different percentages of fossil fuel displacement. Only symbols for 100% fossil fuel replacement are reported for blue $H_2$ with 1% $CH_4$ leakage. Also reported is the difference in $CO_2$ concentration ($\Delta[CO_2e]$) that would produce equivalent radiative forcing to the change in equilibrium $CH_4$ (upper axis).

hydrogen production pathway. For green $H_2$, i.e., hydrogen obtained from renewable sources, we scale $CH_4$ emissions based on the reduced consumption of fossil fuels resulting from hydrogen usage (Fig. 2b). Estimates of current methane emissions associated with fossil fuel extraction and distribution are in the range 80–160 Tg $CH_4$ $yr^{-1}$[18,24,25] and relatively equally distributed among coal, oil, and gas sectors[26]. Here we use the top-down estimate of 111 Tg/year[18].

For blue $H_2$, which is derived from steam methane reforming (SMR), the variation in $CH_4$ sources not only accounts for the reduced consumption of fossil fuels but also for the methane emissions (venting, incomplete combustion, fugitive) associated with blue hydrogen production. These emissions depend on the amount of $CH_4$ needed to produce $H_2$, i.e., feedstock and energy requirements of the SMR process ("Methods"), and the $CH_4$ leak rate. The precise average leak rate of the global natural gas supply chain remains uncertain. One of the reasons is that national inventories generally underestimate real emissions[27–30]. More detailed studies relying on field measurements in the United States and Canada estimate average leak rates around 2%[28–30], with large spatial heterogeneity between different operators[31]. Although national inventories suggest that some countries, like Venezuela and Turkmenistan, have higher leak rates[26], here we adopt 2% as the maximum global $CH_4$ leak rate for our scenarios, because methane-mitigation efforts are likely to decrease future global leak rates[21] and, more importantly, because not all hydrogen produced will be blue $H_2$. In this regard, the scenario of blue $H_2$ with a 2% $CH_4$ leak rate can also be interpreted as a combination of equal production of

green $H_2$ and blue $H_2$ with 4% $CH_4$ leak rate. We use 0.2% as a lower bound for the $CH_4$ leak rate, since this has been declared as the target of several energy companies for 2025[32]. 1% represents an intermediate scenario of blue $H_2$ production.

Figure 2b shows the resulting $CH_4$ emissions associated with green and blue $H_2$ production with methane leak rates of 0.2, 1, and 2%. The different leak rates have a great impact on the methane emissions. Compared to the fossil fuel energy system, $CH_4$ emissions are reduced in the blue $H_2$ scenario with 0.2% methane losses, but largely increased in the blue $H_2$ scenario with 2% methane losses. The fossil fuel displacement by blue $H_2$ with 1% methane losses shows basically no net effect on the $CH_4$ emissions.

As a specific case, we also investigate the $H_2$ and $CH_4$ emission changes associated with estimates of future hydrogen production in a set of net-zero scenarios. $H_2$ production is expected to increase from current 90 Tg/year to 530–660 Tg/year in 2050[2,33,34]. We thus consider a 500 Tg/year rise in the global $H_2$ production, which is energetically equivalent to about 15% of current fossil fuel energy. Figure 3a shows how, depending on the $H_2$ production pathway and the different hydrogen and methane leak rates, the emission changes of these two gases can vary substantially.

## Tropospheric response

For the previous emission scenarios, we evaluate the changes in the equilibrium concentrations of tropospheric hydrogen and methane, namely $\Delta[H_2]$ and $\Delta[CH_4]$. The timescales to equilibrium are dictated

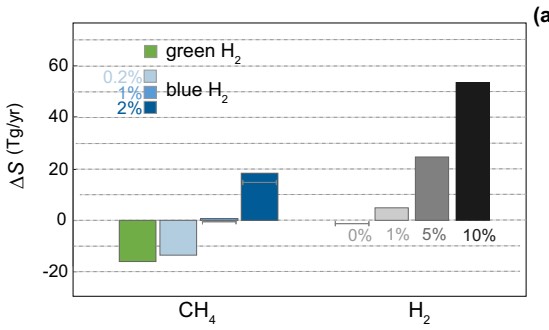
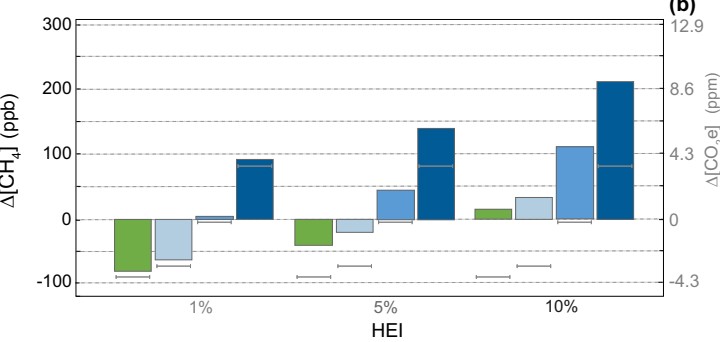

Fig. 3 | **Methane response to increasing $H_2$ production. a** Changes in $H_2$ and $CH_4$ sources ($\Delta S$) due to green and blue $H_2$ production ($\approx$500 Tg yr$^{-1}$). HEI is the $H_2$ emission intensity. Gray lines mark the case for HEI = 0%. Blue bars for $\Delta S_{CH_4}$ are obtained with HEI = 10%. **b** Response of $CH_4$ atmospheric concentration. The right axis shows the $\Delta[CO_2e]$ that would produce equivalent radiative forcing to the change in equilibrium $CH_4$.

by the gas average lifetimes ("Methods"). The corresponding variations in steady state concentration of OH are reported in Supplementary Figs. 1 and 2.

The $H_2$ economy causes a rise in tropospheric $H_2$ as a result of the additional emissions (Fig. 2c). The intensity of this increase varies considerably as a function of the emissions of the hydrogen value chain. The concentration variation could go from less than 100 ppb$_v$ to more than 2000 ppb in envisioned scenarios of the $H_2$ economy, namely a +300% from the current $H_2$ tropospheric level.

The response of atmospheric $CH_4$ results from the combination of the methane emission change and the methane sink weakening due to the higher hydrogen emissions. To discriminate between the two mechanisms, it is useful to focus on the scenarios of fossil fuel displacement by green $H_2$. In the case of a perfectly sealed green $H_2$ value chain (HEI = 0%), [$CH_4$] and [$H_2$] both decrease due to reduction in fossil fuel emissions. As $H_2$ emissions increase (HEI > 0), $\Delta[CH_4]$ increases too. Up to the point that when HEI overcomes a critical threshold, there is an increase in atmospheric methane, i.e., $\Delta[CH_4] > 0$, even though methane emissions are lower. This critical HEI is in the range 8–10% for green $H_2$ as it has a weak nonlinear dependence on the percentage of fossil fuel energy that is replaced by $H_2$ (see also Supplementary Fig. 3).

The scenarios of blue $H_2$ with 0.2% $CH_4$ leak rates are not very different from the green $H_2$ scenarios, with the critical HEI being in the range 7–8%. Regarding the scenarios of blue $H_2$ with 1% $CH_4$ leak rates, since there is basically no change in the methane emissions (Fig. 2b), the methane response is only associated with the reduction in OH availability due to the higher $H_2$ concentration. The critical HEI is not defined for this blue $H_2$ as the methane burden increases in all cases. The worst scenarios of blue $H_2$ with 2% $CH_4$ leak rates show drastic differences in the tropospheric concentrations of the two gases, which increase considerably, with a weakly nonlinear effect due to the drop in atmospheric OH.

The atmospheric methane response to future $H_2$ production[2,33,34] shows qualitatively similar results as a function of the $H_2$ production pathway and the percentage of $H_2$ lost to the atmosphere (Fig. 3b). Positive effects in terms of methane mitigation are observed only for green and blue $H_2$ with low methane losses, if the $H_2$ emission intensity is well below 10%. Otherwise, the tropospheric methane burden is enhanced.

We also evaluated the change in $CO_2$ concentration ($\Delta[CO_2e]$) that would produce equivalent radiative forcing to the change in the equilibrium concentration of $CH_4$ (Figs. 2c and 3b). We used the radiative efficiency of $CH_4$ that includes indirect effects on $O_3$ and stratospheric $H_2O$[35]. Under the worst scenario of blue $H_2$ production with 2% $CH_4$ losses and 10% $H_2$ losses, the rise in equilibrium $CH_4$ due to future $H_2$ production would be like adding 9 ppm of $CO_2$ to the

atmosphere (Fig. 3b). For the same blue $H_2$, the rise in $CH_4$ following the entire displacement of fossil fuels would be like adding around 70 ppm of $CO_2$ (Fig. 2c). This is equivalent to around 50% of the $CO_2$ increase from preindustrial times (278 ppm) to current days (417 ppm). Since the goal of keeping the global average temperature rise below 1.5 °C requires a mid-century maximum of $CO_2$ close to 450 ppm, these results support previous concerns about the sustainability of blue $H_2$[36] unless fugitive emissions can be kept sufficiently low.

### Critical HEI for methane mitigation

The quantification of the critical hydrogen emission intensity (HEI$_{cr}$) for methane mitigation is key to assess whether displacing fossil fuels with hydrogen would mitigate or enhance the tropospheric burden of $CH_4$. Here we investigate how the HEI$_{cr}$ is affected by the hydrogen production pathway and by two of the most uncertain terms in the $CH_4$-$H_2$-OH balance: (i) the partitioning of the OH sink among the tropospheric gases; (ii) the rate of $H_2$ uptake by soil bacteria. The derivation of an analytical solution for the HEI$_{cr}$ is reported in the "Methods".

The very short lifetime of OH makes the quantification of its atmospheric dynamics extremely challenging. Indirect methods are typically used to estimate OH concentrations, sources, and sink partitioning[37–39]. Using a range of OH partition estimates[38,40], we investigate the dependence of the HEI$_{cr}$ to different values of OH excess ($E_{OH}$), $E_{OH}$ being the excess of OH that is consumed by other tropospheric gases besides hydrogen, methane, and carbon monoxide. Figure 4 shows the quasi-linear response of the HEI$_{cr}$ to $E_{OH}$. We stress that a variation in $E_{OH}$ is equivalent to a variation in the OH sources since we preserve the current average OH concentration, which is relatively well constrained by inverse modeling[37,41].

The HEI$_{cr}$ is much lower for blue $H_2$ than for green $H_2$ because of the methane emissions associated with blue $H_2$ production. For the current tropospheric conditions, we find that HEI$_{cr}$ is around 9% for green $H_2$, around 7% for blue $H_2$ with 0.2% methane leak rates, and 4.5% for blue $H_2$ with 0.5% methane leak rates. Blue $H_2$ with 1% methane leak rate has a HEI$_{cr}$ that is close to zero, as displacement of fossil fuel with this hydrogen does not reduce methane emissions (Fig. 3b). For even higher methane leak rates, the methane burden would increase regardless of the $H_2$ emissions, so that the HEI$_{cr}$ is negative.

The $H_2$ uptake by soil bacteria is another crucial process in the evaluation of HEI$_{cr}$ and in the overall $CH_4$–$H_2$–OH dynamics, since it accounts for 70–80% of $H_2$ tropospheric removal[11]. Despite recent research on uptake modeling[42,43] and the microbial characterization of the $H_2$-oxidizing bacteria[44], the spatial heterogeneity of the uptake as driven by local hydro-climatic and biotic conditions hinders bottom-up estimates of the global average uptake rate. In atmospheric studies, the average uptake rate is usually adjusted in order to obtain a reasonable simulation of observed surface hydrogen concentrations[14,45].

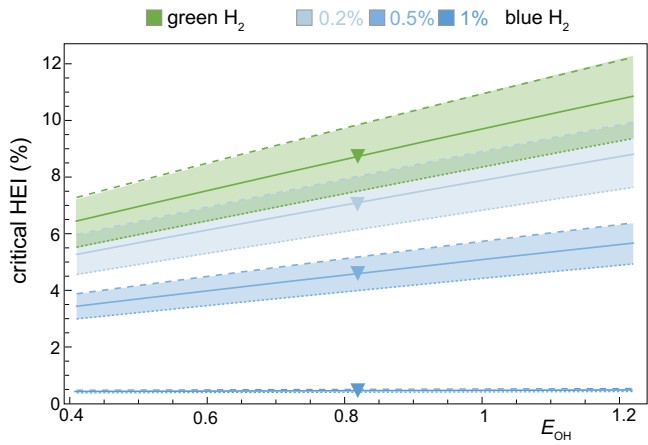

**Fig. 4 | Critical hydrogen emission intensity (HEI) for methane mitigation.**
Critical HEI as a function of OH excess ($E_{OH}$) and hydrogen production method
(green and blue $H_2$ with 0.2, 0.5, 1% $CH_4$ leak rates, respectively). Dashed (dotted)
lines are obtained for a 20% increase (decrease) in the $H_2$ uptake rate by soil bacteria ($k_d$). Triangles mark the critical HEI for the best estimate of $E_{OH}$.

To account for these potential sources of uncertainties, we show how a
±20% variation in the uptake rate influences the critical HEI (bands in
Fig. 4). A stronger biotic sink (dashed lines) reduces the consumption
of OH by $H_2$ and, consequently, increases the $HEI_{cr}$. A weaker biotic
sink (dotted line) has the opposite effect.

Regarding the impact of climate change on the $H_2$ soil sink, recent
studies indicate that increasing temperatures are expected to slightly
favor the uptake on a global scale[14], while shifts in rainfall regimes will
be the significant drivers of $H_2$ uptake changes at the local scale[43]. From
a biotic perspective, the adaptability of $H_2$-oxidizing bacteria to
extreme environments[46] suggests that their presence will remain
widespread in the future, but their spatial heterogeneity may change as
a result of climate and anthropogenic pressures.

Another source of uncertainties in the evaluation of $HEI_{cr}$ is related to the estimate of $CH_4$ emissions associated with fossil fuel use.
Since there is a quasi linear relationship between these emissions and
the $HEI_{cr}$ (Eq. (16) in "Methods"), the same relative uncertainty of fossil
fuel methane emissions (Fig. 1) applies to the $HEI_{cr}$.

## Discussion

The success of the global net-zero transition hinges on hydrogen as a
scalable low-carbon energy carrier that can replace fossil fuels in several hard-to-electrify energy and transport sectors. More than 20
governments and many companies have already announced strategies
for hydrogen production, and the numbers are likely to increase as
policy frameworks that facilitate hydrogen adoption are promoted[1,2].
Considerable investments are still needed to achieve such a transition,
as the current hydrogen momentum falls short compared to net-zero
goals. The Hydrogen Council[2] estimates that there is a USD 540 billion
gap between the investments of announced projects (USD 160 billion)
on hydrogen production and the investments required by 2030 to be
on a net-zero pathway (USD 700 billion).

While the positive effects of a more hydrogen-based economy are
relatively established (e.g., lower $CO_2$ emissions, decreased urban
pollution, etc.), considerable uncertainty still surrounds the consequences of hydrogen emissions to the atmosphere, because of
potential indirect GHG effects[14,19]. Here we have focused on the impact
of a more hydrogen-based energy system on tropospheric methane,
the second most important greenhouse gas.

We have shown how the replacement of fossil fuel energy with
green or blue hydrogen could have very different consequences for
tropospheric $CH_4$, depending on the amount of hydrogen lost to the

atmosphere and the methane emissions associated with hydrogen
production (Figs. 2 and 3). Specifically, tropospheric $CH_4$ would
decrease due to the fossil fuel displacement only if the rate of $H_2$ losses
is kept below the critical HEI.

This is around 9 ± 3% for green $H_2$ (Fig. 4). The same critical value
would apply to other $H_2$ colors that do not entail the use of fossil fuels,
like white or orange $H_2$ extracted from underground deposits[12,47]. The
critical HEI for blue $H_2$ is much lower due to the $CH_4$ emission associated with blue $H_2$ production. We have found that the methane
emissions in a blue $H_2$ economy could be higher than in a fossil fuel
economy if the methane supply chain had an average leak rate above
1%. Furthermore, the superimposition of $CH_4$ and $H_2$ emissions may
have undesired consequences for the tropospheric burden of $CH_4$.
This may be a potential problem in the near term, given that steam
methane reforming will be used to bridge the gap between increasing
$H_2$ demand and limited green $H_2$ production capacities[2]. Our results
suggest that including hydrogen emissions would aggravate the
greenhouse gas footprint of blue $H_2$[36].

In addition to the $CH_4$ feedback, $H_2$ emissions are also expected to
impact ozone ($O_3$) and stratospheric water vapor ($H_2O$), with negative
consequences for both air quality and radiative forcing. Accounting for
these effects, we can provide a comparison between the radiative
forcing of hydrogen-based and fossil fuel-based energy systems.
Because both $H_2$ and $CH_4$ are short-lived gas compared to $CO_2$, the
time horizon for this comparison is crucial[48]. Here we consider 20-year
and 100-year time horizons. The GWP of $H_2$ is estimated at 11 ± 5 (100-
year) and $33^{+11}_{-13}$ (20-year)[16]. The GWP of $CH_4$ is estimated at 28 (100-
year) and 80 (20-year)[35]. In an envisioned hydrogen economy that
replaces the current fossil fuel industry, the $H_2$ emissions could be in
the range 23 to 370 Tg $H_2$ yr$^{-1}$, for a $H_2$ emission intensity going from 1
to 10% (Fig. 2a). These emissions would have a radiative forcing impact
of 0.7–12% (100-year) and 2–35% (20-year) of the current $CO_2$ emissions from fossil fuels ($\approx$35 Pg $CO_2$ yr$^{-1}$). If the global $H_2$ economy relied
on blue $H_2$ with a 2% methane leakage rate, methane emissions would
cause an additional radiative forcing impact that is around 10% (100-
year) and 27% (20-year) of the current $CO_2$ emissions from fossil fuels.
Hence, in the worst scenario, up to 22% of the climate benefits of the
hydrogen economy could be offset by gas losses over a 100-year
horizon. The percentage could be as large as 65% over a 20-year horizon. These values could be higher on a regional scale if the leak rate of
the natural gas supply chain is above 2%.

To maximize the climate benefit of hydrogen adoption, minimizing both $H_2$ and $CH_4$ losses across the supply chain of hydrogen
production will need to be a priority. On the methane side, some
governments and companies have already committed to reducing the
leaks from the oil and gas sector, because this could be the most cost-
effective and impactful action for near-term climate mitigation[21]. The
International Energy Agency (IEA) estimates that, with the recent rise in
natural gas prices, the abatement of methane emissions from the
global gas and oil sector could be implemented at no net cost[49]. Hence,
the accomplishment of this mitigation is only a matter of political will
for the limited number of companies involved.

On the hydrogen side, the global value chain still has to be built.
This offers the advantage of tackling the hydrogen emission problem
ahead of time. On the one hand, energy companies will have a great
interest in minimizing economic loss and safety risks due to hydrogen
leaks. On the other hand, however, many technological challenges still
need to be addressed. First, $H_2$ containment may remain an issue even
as technologies progress. The high diffusivity of the small $H_2$ molecule
has already challenged the scientific community's ability to measure
the $H_2$ concentration in the atmosphere[50] and in the firn air of ice
sheets[51]. Second, while more field-based estimates of $H_2$ losses are
needed, there is currently no commercially available sensing technology able to detect small $H_2$ leaks at the ppb level[48]. Third, global-space
monitoring, which is bringing a much-needed transparency to the

quantification of real methane emissions[27,31], will also require new technology since $H_2$, unlike $CH_4$ or $CO_2$, does not absorb infrared radiation. For all these reasons, the uncertainty about future emissions from the $H_2$ value chain remains large.

Our versatile atmospheric model allowed a broad exploration of scenarios in a hydrogen-based energy system. Simulations with high resolution three-dimensional atmospheric chemistry models, which are more comprehensive but more computationally demanding, could refine our results for specific scenarios. In particular, a more detailed model could improve the assessment of $H_2$ displacement of fossil fuels by accounting for the emission changes of other chemical species, like CO and $NO_x$, which impact the $CH_4$–$H_2$–OH dynamics. Further analyses could also refine the potential changes in emission inventories due to $H_2$ displacement of different fossil fuels.

## Methods

### The Model

With the increasing anthropogenic alteration of atmospheric chemistry, detailed three-dimensional atmospheric chemistry models have become critical to evaluate the atmospheric interactions with the climate forcing[52,53]. Nonetheless, thanks to their versatility, simplified models of atmospheric chemistry have also proven very useful to investigate the fundamental processes governing the coupling between atmospheric gases and the consequences of their possible perturbations (e.g., refs. 54–59). The insights obtained with the $CH_4$–CO–OH model by Prather et al.[54], in particular, led to a +40% revision of the IPCC's GWP for $CH_4$[60]. Here we extend Prather's seminal model by adding the mass balance equation for atmospheric $H_2$. The purpose is to identify the key components that control the $H_2$ feedback on the tropospheric dynamics of $CH_4$ (Fig. 1).

The chemical reactions considered are

$$CH_4 + OH \xrightarrow{k_1} \dots \longrightarrow \alpha\, H_2 + CO \dots, \quad R_{CH_4} = k_1[OH][CH_4], \quad (1)$$

$$H_2 + OH \xrightarrow{k_2} \dots, \quad R_{H_2} = k_2[OH][H_2], \quad (2)$$

$$CO + OH \xrightarrow{k_3} \dots, \quad R_{CO} = k_3[OH][CO], \quad (3)$$

$$X + OH \xrightarrow{k_4} \dots, \quad R_X = k_4[OH][X], \quad (4)$$

with $R$ representing the rates of reactions, $[\cdot]$ the concentrations, and $k_i$ the rate coefficients. We indicated only the products with which we are concerned, the CO and $H_2$ produced by oxidation of $CH_4$(1). $H_2$

production through $CH_4$ oxidation has yield $\alpha \approx 0.37$[13]. X encompasses all the other species, besides $CH_4$, CO, and $H_2$, that consume OH. Based on the above reactions, the balance equations for the $CH_4$–$H_2$–CO–OH system are

$$\frac{d[CH_4]}{dt} = S_{CH_4} - R_{CH_4} - R_s, \quad (5)$$

$$\frac{d[H_2]}{dt} = S_{H_2} + \alpha R_{CH_4} - R_{H_2} - R_d, \quad (6)$$

$$\frac{d[CO]}{dt} = S_{CO} + R_{CH_4} - R_{CO}, \quad (7)$$

$$\frac{d[OH]}{dt} = S_{OH} - R_{CH_4} - R_{H_2} - R_{CO} - R_X, \quad (8)$$

where $R_d = k_d[H_2]$ is the $H_2$ uptake by soil bacteria, which plays a crucial role in the global balance of $H_2$ since it accounts for around 70–80% of tropospheric removal[11,43,61]; $R_s = k_s[CH_4]$ accounts for the smaller sinks of $CH_4$, namely soil uptake, stratospheric loss and reactions with chlorine radicals[62]. For simplicity, we neglect the smaller sinks of $H_2$, i.e., stratospheric loss (≈1% of removal[63]), and CO, i.e., soil uptake and stratospheric loss (<10% of removal[64]).

The solution at quasi steady state (i.e., $d[\cdot]/dt = 0$) provides the sources for fixed tropospheric concentrations. Positive solutions for OH occurs if $S_{OH} > (2 + \alpha)(S_{CH_4} - R_s) + S_{CO} + S_{H_2} - R_d$, i.e., when there is enough OH to oxidize all CO sources, the part of $CH_4$ sources that is not balanced by smaller sinks, and the part of $H_2$ sources that is not balanced by the soil uptake. The excess of OH consumed by other gases, besides $CH_4$, CO, and $H_2$, can be defined as $E_{OH} = R_X/(R_{CH_4} + R_{CO} + R_{H_2})$. The values representing average tropospheric conditions are summarized in Table 1. The values of $S_{OH}$ and $S_{CO}$ are kept constant in all scenarios.

### Linear stability and transient dynamics

We investigate the effects of an emission pulse of $H_2$ on the tropospheric system (5)–(8). The timescales and modes of the atmospheric response to chemical perturbations are defined by the eigenvalues and eigenvectors of the system[54,55]. Indicating with $\mathbf{c}(t)$ the solution vector of the system (5)–(8), the temporal dynamics of a small perturbation $\hat{\mathbf{c}}$ around $\mathbf{c}$ evolves as

$$\frac{d\hat{\mathbf{c}}}{dt} = \mathbf{J}\hat{\mathbf{c}}, \quad (9)$$

**Table 1 | Tropospheric budgets of key species and definition of linear stability modes**

|  |  | $CH_4$ | $H_2$ | CO | OH | $-\lambda_i^{-1}$ (yr) |
|---|---|---|---|---|---|---|
| Steady state | Concentration (ppb) | 1890 | 530 | 80 | $10^6$ cm$^{-3}$ |  |
|  | Sources (ppb/yr) | 226 | 265[a] | 480[a] | 1333 |  |
|  | $\tau$ (yr) | 8.3 | 2 | 0.17 | 1 s |  |
| Linear stability | $CH_4$ mode | 1% | 0.31% | 0.64% | −0.39% | 12.3 |
|  | $H_2$ mode | −0.01% | 1% | 0.03% | −0.06% | 2 |
|  | CO mode | −0.008% | 0.001% | 1% | −0.36% | 0.2 |
|  | OH mode | …[b] | …[b] | …[b] | 1% | 1.5 s |

[a]Sources for CO and $H_2$ include production from $CH_4$ oxidation.

[b]… is <10$^{-7}$.

Sources are obtained from the system (5)–(8) at steady state with the current tropospheric concentrations. $\tau$ is the average lifetime of each gas. The modes are expressed as relative changes normalized so that the dominant species' ratio is 1%. Reaction rates are defined as follows: $k_1 = 3.17 \times 10^{-15}$ cm$^3$/s; $k_2 = 3.8 \times 10^{-15}$ cm$^3$/s; $k_3 = 1.9 \times 10^{-13}$ cm$^3$/s; $k_s = 0.02$ yr$^{-1}$; $k_d = 0.38$ yr$^{-1}$ is such that soil uptake accounts for 75% of atmospheric $H_2$ removal; $k_4[X] = 0.3$ s$^{-1}$ ($E_{OH} = 0.82$) is defined so that 45% of OH is consumed by the species X, 36% by CO, 14% by $CH_4$, and 5% by $H_2$[38]. Concentrations are converted to mixing ratios using 1 ppb = $1.57 \times 10^{10}$ cm$^{-3}$; sources are converted from ppb/yr to Tg/yr using $4.22 \times 10^{18}$ kg as the troposphere mass[68].

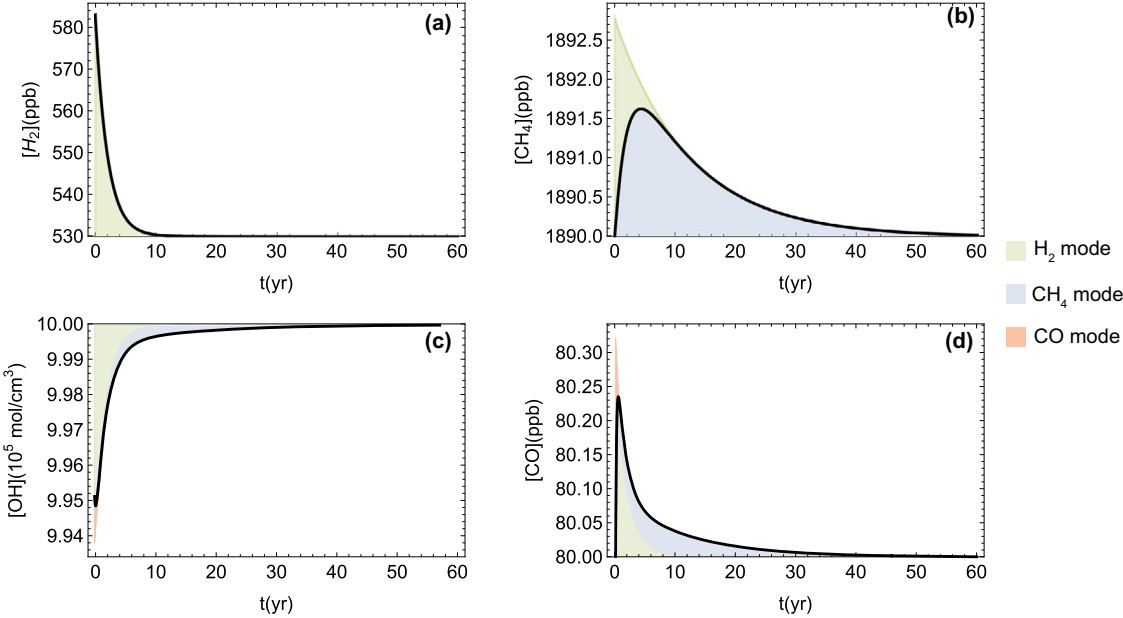

**Fig. 5 | Transient dynamics.** Tropospheric response to a pulse of $H_2$ (10% increase of its concentration). Temporal dynamics of $H_2$ (**a**), $CH_4$ (**b**), OH (**c**), and CO (**d**). Colors highlight the contributions of the different modes. When different modes superimpose, the faster-decaying mode is shown on top of the others.

where **J** is the Jacobian of the system evaluated in **c**. For the equilibrium solution $\mathbf{c}_0$ representing the current tropospheric concentrations, the eigenvalues and eigenvectors, or modes, of the linearized system (9) are reported in Table 1. Since all eigenvalues are real and negative ($\lambda_i < 0$), the equilibrium solution $\mathbf{c}_0$ is a stable node. As a result, any small perturbation asymptotically decays in time with a timescale defined by the negative reciprocal of the eigenvalue.

Because the system equations are coupled, the decay timescale ($-\lambda_i^{-1}$) of a gas perturbation does not necessarily correspond to the gas steady state average lifetime ($\tau_i$). The $CH_4$ perturbation, in particular, decays with a timescale that is much larger than what predicted by its steady state lifetime, i.e., $R = -\lambda_{CH_4}^{-1}/\tau_{CH_4} > 1$. This mechanism, known as the $CH_4$ feedback effect[55,65], has a crucial role in increasing the GWP and the environmental impact of $CH_4$ emissions. Detailed models of atmospheric chemistry usually provide $R$ around 1.3–1.4[65]. We find a marginally higher feedback factor, namely $R \approx 1.5$, in agreement with previous findings using Prather's box model[54,55,57]. The decay timescale of the $H_2$ perturbation instead corresponds to the $H_2$ average lifetime, namely $-\lambda_{H_2}^{-1} \approx \tau_{H_2}$, in agreement with results from detailed atmospheric chemistry models[19].

While the modal eigenvalue analysis correctly captures the asymptotic stability of the solution $\mathbf{c}_0$, it does not describe the perturbation dynamics at finite times, i.e., before the asymptotic decay. Still within the domain of the linearized system (9), a more complete picture can be obtained by analyzing the temporal evolution of the solutions with specific attention to the emergence of transient growth phenomena, which are known to occur in systems where the modes are non-orthogonal, as in the present case. When large enough, a transient growth can even trigger nonlinearities that destabilize the equilibrium solution[66].

Figure 5 shows the transient growth phase of tropospheric $CH_4$ and CO that follows a 10% perturbation of $H_2$ concentration. Specifically, the pulse of $H_2$ causes a drop in OH and a build-up of $CH_4$ that lasts a few years, while the $H_2$ perturbation decays with the timescale $\tau_{H_2}$. The $CH_4$ build-up then decays in the same manner as would a direct pulse of $CH_4$ with a timescale defined by the $CH_4$ feedback effect. In analytical terms, the perturbation of tropospheric $CH_4$ mainly

due to the excitation of $H_2$ and $CH_4$ modes is given by $\delta[CH_4] \approx 2.76e^{\lambda_{CH_4}t} - 2.82e^{\lambda_{H_2}t} + 0.06e^{\lambda_{CO}t}$.

Using this result in traditional GWP formulas[35] yields a GWP for $H_2$ due to direct $CH_4$ perturbation around 7.8 with the 100-year time-horizon and 22 with the 20-year time horizon. It is estimated that around half of the $H_2$ indirect radiative forcing is due to the direct $CH_4$ perturbation, and the other half to the $O_3$ and stratospheric $H_2O$ impacts caused by both $H_2$ and $H_2$-induced $CH_4$ perturbations[14]. Taking this into account yields a total GWP for $H_2$ of 15.6 with the 100-year time-horizon and 44 with the 20-year time-horizon. These values are in the upper range of the recent estimates of $11 \pm 5$ for GWP100 and $33^{+11}_{-13}$ for GWP20 obtained with a detailed model of atmospheric chemistry[16]. Notably, the consequences of the $H_2$ pulse on $CH_4$ are relatively small in magnitude because most of the additional $H_2$ is oxidized by soil bacteria and not by OH. The stability of this biotic sink as affected by climate change and anthropic pressure is hence a crucial aspect for the impact of future $H_2$ emissions, as further discussed in the main text.

## Critical hydrogen emission intensity

We here derive an explicit expression for the critical $H_2$ emission intensity (HEI$_{cr}$) for methane mitigation, defined as the emission rate that offsets the $H_2$ replacement of fossil fuels. The expression is derived for an infinitesimal replacement of fossil fuel energy with $H_2$ ($dE$ in ExJ/yr), but well approximates the critical HEI for finite replacement of fossil fuel energy (see Supplementary Fig. 3). As a first step, we differentiate the system (5)–(8) at equilibrium ($d[\cdot]/dt = 0$) with respect to $E$. This yields

$$S_{CH_4,E} - k_1[CH_4][OH]_E = 0, \tag{10}$$

$$S_{H_2,E} + \alpha k_1[CH_4][OH]_E - k_2([H_2][OH])_E - k_d[H_2]_E = 0, \tag{11}$$

$$k_1[CH_4][OH]_E - k_3([CO][OH])_E = 0, \tag{12}$$

$$k_1[CH_4][OH]_E + k_2([H_2][OH])_E + k_3([CO][OH])_E + k_4[X][OH]_E = 0, \tag{13}$$

where subscript $E$ indicates $d \cdot /dE$. $[CH_4]_E = 0$ because of the definition of the critical $H_2$ emission intensity, which leaves the methane concentration unaltered. We consider that only $H_2$ and $CH_4$ sources vary with $E$, while $S_{OH,E} = S_{CO,E} = 0$. These variations can be estimated as

$$S_{H_2,E} = a_{H_2}\left(-ff_{H_2} + \frac{HEI}{\eta_{H_2}(1-HEI)}\right), \quad (14)$$

$$S_{CH_4,E} = a_{CH_4}\left(-ff_{CH_4} + \frac{r\, MEI}{\eta_{H_2}(1-HEI)}\right), \quad (15)$$

where HEI and MEI are the hydrogen and methane emission intensities, respectively (MEI = 0 for green $H_2$); $\eta_{H_2}$ is $H_2$ higher heating value; $r$ is the amount of $CH_4$ needed to produce a unit of blue $H_2$; $ff_{CH_4}$ and $ff_{H_2}$ are the average amounts of $CH_4$ and $H_2$ emitted per ExJ of fossil fuel energy; $a_{H_2}$ and $a_{CH_4}$ are conversion factors.

Substituting Eqs. (14), (15) into the system (10)–(13) and after some algebra, one obtains the critical $H_2$ emission intensity

$$HEI_{cr} = \frac{A\left(ff_{CH_4}\,\eta_{H_2} - r\, MEI\right) + B\, ff_{H_2}\,\eta_{H_2}}{A\, ff_{CH_4}\,\eta_{H_2} + B\left(ff_{H_2}\,\eta_{H_2} + 1\right)}. \quad (16)$$

where the dependence to the atmospheric composition is embedded in $A = k_d(k_4[X] + k_2[H_2] + 2k_1[CH_4]) + k_2[OH]((\alpha+2)k_1[CH_4] + k_4[X])$ and $B = 8k_1k_2[CH_4][OH]$. Parameters have been defined as follows: $\eta_{H_2} = 0.143$ ExJ/Tg$_{H_2}$, $r = 3.2$ kg$_{CH_4}$/kg$_{H_2}$, $ff_{CH_4} = 0.225$ Tg$_{CH_4}$/ExJ, $ff_{H_2} = 0.0225$ Tg$_{H_2}$/ExJ, $a_{CH_4} = 0.43$ ppb/Tg, $a_{H_2} = 8a_{CH_4}$. To obtain the value of $r$, we used the estimate of 3.7 kg of natural gas for kg of $H_2$[67], which includes feedstock and energy requirements, and we assumed that 85% of natural gas by weight is composed by methane. $ff_{CH_4}$ and $ff_{H_2}$ are obtained as the ratio between the global $CH_4$ and $H_2$ emissions due to fossil fuel use and the global fossil fuel energy.

## Data availability
All data generated during this study are provided in the supplementary dataset file.

## Code availability
The code used to generate the results is provided in the supplementary dataset file.

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

## Acknowledgements

We acknowledge support from the US National Science Foundation (NSF) grant nos. EAR1331846 and EAR-1338694, the BP through the Carbon Mitigation Initiative (CMI) at Princeton University, and the Moore Foundation. We thank Larry Horowitz for critical reading of the manuscript.

## Author contributions

M.B.B., S.W.P., and A.P. conceptualized the work. M.B.B. developed the analytical model with contributions from F.P., analyzed the results and prepared the manuscript. A.P., F.P., and S.W.P. supervised the work and edited the manuscript.

## Competing interests

The authors declare no competing interests.
