## [Peer Review File · Nature Communications]

Risk of The Hydrogen Economy For Atmospheric MethaneReviewer #1 (Remarks to the Author):

The manuscript titled 'Tropospheric Methane Enhancement by Hydrogen Fugitive Emissions' presents the results of modelling the effect of H₂ leakages on methane enhancement. The work estimates a critical leakage rate of H₂, the point at which displacement over fossil fuels bears no climate benefit. The work follows on from recent works which estimate the climate impacts of H₂ emissions and provides an interesting perspective on the H₂ and methane budgets. While the work is interesting and topical given the high interest in both methane and H₂, the paper is not suitable for publication in Nature in its current state. The underlying leakage assumptions and data are not rigorous which is the main weakness of the paper. I outline this concern, and other in more detail below.

- H₂ leakage rate: it is unclear whether the authors only consider leaks of H₂ i.e. emissions from leaky equipment/processes, so excluding other emissions such as venting and other fugitives (irregular/malfunctioning equipment).
- H₂ leakage rate: there is a very high level of uncertainty in what the H₂ emissions could be in potential future H₂ supply chains. It is unclear in the manuscript if the authors only consider H₂ emissions from production or consider entire supply chains. The authors use literature values to derive a range of 1-10% but given the uncertainty, and that it is not specified what parts or processes of the H₂ supply chain this covers, this is not appropriate and the authors should have attempted to model their own H₂ leakage rates. There are also more recent studies which estimate H₂ emissions from H₂ supply chains and values from these could have been used or their methods applied: <https://www.sciencedirect.com/science/article/pii/S004896972201717X> or <https://www.gov.uk/government/publications/fugitive-hydrogen-emissions-in-a-future-hydrogen-economy>
- Methane leakage rate: it is unclear which sections of the gas supply chain are included in the emission rates assumed. For blue H₂, the stages of production, gathering and boosting, processing and transmission would be what would typically be considered but this varies depending on how far away the SMR plant is from the gas field, the size of the gas field and the quality of the gas produced in the gas field.
- Methane leakage rate: the methane leakage rates assumed (0.2 to 3%, average of 1.5%) is unclear if the authors are considering the US only or a global scale. It is also unclear if only methane leakages are considered and if other methane fugitives, venting and incomplete combustions emissions are excluded. The methane emission rate varies significantly from country to country so if estimating on a global context, it is not appropriate to use values specific to a handful of countries. Also, it is confusing whether coal is considered as blue H₂ is typically natural gas SMR with CCS- referring to page 4 lines 107 to 122.
- Methane leakage rate: the 3% assumed is from the paper by Howarth and Jacobson in their blue H₂ paper. It is unclear in the Howarth and Jacobson paper where this 3% comes from and what it is representative of. In the paper they cite the 3.5% emissions to be based on top-down estimates for emissions from 20 different studies in 10 different gas fields plus a top-down estimate for emissions from gas transport and storage. However, when the citation for this value is checked the reference given is for a book which is not yet published so it is uncertain what the 3/3.5% is representative of.
- H₂ production: the projection by the Hydrogen Council was used but there are other studies which estimate future H₂ demand, such as the IEA and IPCC. Why were these not used and a range in H₂ production not considered?
- The paper would benefit from placing their results in context of impact on meeting climate goals. This would broaden the applicability of the results of the work and who would benefit from the work. In the Methods, a H₂ GWP₁₀₀ was estimated, which could have been applied to the estimated H₂ emissions. Also, given that H₂ perpetuates the impacts of methane, could a GWP for methane not have been calculated which takes into account the effect of a prolonged atmospheric lifespan?
- The authors considered a pulse of H₂ emissions. In the recent paper by Okko and Hamburg, they argue that this is not appropriate for H₂ given the likely emissions profile of emissions from H₂ supply chains. They argue continuous emissions rather than a pulse would better reflect the climate impacts of H₂.

- The paper has grammatical mistakes and typos throughout, and sections are difficult to read. Please have the paper proofread by a native English speaker.
- Figure 1: could an average value be added to the flux estimates as some of the min-max ranges are large?
- Page 4 line 93-94: 'or increase ($\Delta\text{SCH}_4 > 0$), due to an additional methane demand for blue H₂ production' this is unlikely given initiatives to cut methane emissions through efforts to improve emissions detection e.g. OGMP2.0, EU methane regulation. Also, if traditional demands for natural gas dropped but demand for blue H₂ increased, would this result in $\Delta\text{SCH}_4 > 0$?

Reviewer #2 (Remarks to the Author):

Review of 'Tropospheric Methane Enhancement by Hydrogen Fugitive Emissions' by Bertagni et al.

This paper uses a simple model to investigate the response of atmospheric methane to H₂ leakage from a potential future hydrogen energy industry. The impact of H₂ leakage on stratospheric water vapour and tropospheric ozone, which has been estimated by more complex models with detailed chemistry to represent about 50% of the indirect radiative forcing from hydrogen, is excluded from the analysis. Despite focussing only on the impact of H₂ on methane, the use of the simple model presented here has the advantage that it is possible to explore more of the parameter space within the significant certainties associated with future leakage rates and the present day budget of atmospheric hydrogen.

This study describes a method for deriving a critical leakage rate of H₂ in terms of the impact on methane abundance, taking into account the reduction in methane emissions through displacement of fossil fuels by hydrogen and the chemical feedback of H₂ leakage on the methane lifetime. The results are timely and policy-relevant given the current interest in more widespread use of H₂ as part of commitments to reach net zero.

I believe the methodology is sound, and the paper is well structured and written and I would recommend for publication after the following minor comments are addressed:

I think it could be worth highlighting in the discussion that additional impacts of H₂ leakage on O₃ and H₂O could have further negative effects on the atmosphere, in terms of both air quality and radiative forcing, that are not considered. Also not included, is the impact of displacement of the fossil fuel industry on CO emissions. With reduced fossil fuel usage, CO emissions are also likely to reduce, which will feedback on the CH₄ abundance. Perhaps also a mention that any changes in nitrogen oxide emissions (either increases or decreases) will also influence OH, and thus the ultimate methane abundance in a future hydrogen economy.

Line 18: typo, should read 'not only IS the CLR much lower'

Line 28: If H₂ is combusted, NO_x emissions may increase rather than be reduced relative to the combustion of fossil fuel (e.g. Lewis et al. 2021).

Lines 96-103: In the present day H₂ budget, the fossil fuel industry is estimated to account for approx. 25% of the total H₂ source. It's not completely clear whether the change in H₂ emissions estimated in this study includes a reduction in H₂ emissions from the fossil fuel industry displaced by H₂, as well as the additional H₂ from leakage in an H₂ energy industry – or if it is only the additional H₂ leakage considered. Could the authors clarify?

Lines 108-109: Current global CH₄ emissions from the fossil fuel industry are uncertain (e.g. Schwietzke et al. 2016) – could be worth pointing out this will be another source of uncertainty in the derived CLRs.

Lines 111-122: Could the authors clarify whether the energy efficiency of the CH₄ to H₂ SMR process is taken into account when determining the amount of CH₄ required to generate the H₂ in the 'blue' scenarios? And if so, what energy efficiency is assumed?

Lines 164-172: Presumably these increases in equivalent CO₂ are still much smaller than the CO₂ emission reductions expected from the reduction in size of the fossil fuel industry – is it possible to give some idea of how these numbers compare?

Reviewer #3 (Remarks to the Author):

Bertagni et al provide new insights into the climate implications of shifting from a fossil fuel economy to a hydrogen economy. Their focus on the net impacts on tropospheric methane is innovative and useful. They provide new analyses which help to explore critical tradeoffs that have received limited attention within the literature. The approach is solid, but it is relatively hard work for the reader to get access to the bigger picture characterizing this work and how it fits into the larger issues associated with decarbonization strategies and how effective they are likely to be. The work being reported addresses several of the issues that need to be considered, but not all of them. That is not inherently a problem provided the authors more clearly articulate up front where their work fits into the larger set of issues that need to be considered. The paper would benefit from a broader initial framing and a clearer explanation of the role of the analysis they conduct on informing the decisions that will need to be made and why.

Overall, the paper is well written and would expect it could be of interest to a wide audience if written more accessibly. It is a topic that is of interest to many outside of the scientific community. As currently structured the paper would be an unnecessarily hard read for a non-scientific audience. In order to make it more accessible it would be useful to include some text describing how the CLR is policy relevant and can and cannot be used to understand under what conditions it might make sense to proceed with replacing fossil fuels with hydrogen and the criteria that might be deployed. Such a discussion would help the reader better understand what the CLR could be used for and for what type of questions it would not be relevant. These issues are left unaddressed which will result in confusion on the part of the many, if not most readers.

The paper implicitly assumes that emissions of hydrogen and methane are static, yet there is every reason to believe they are easily malleable given the emerging data on methane emissions, where differences among operators can be huge within the same production basin. I am sure the authors are aware of this, but readers may not. Given that the regulatory requirements/performance standards for methane emissions from natural gas value chains and hydrogen production are currently being debated in many parts of the world, highlighting the need and opportunity to minimize these emissions to capture the decarbonization potential of hydrogen seems important.

Specific comments:

- **The title could be more direct, which would help to orient the reader a bit better i.e. Potential for Tropospheric CH₄ Enhancement with a Hydrogen Economy.**
- **Abstract – description of CLR not clear, clear after reading the paper but not as written in the abstract**
- **Ln 45 – statement that “This indirect radiative forcing is expected to be limited” is too general and not useful. Seems like such a statement is highly subjective and scenario dependent. A more complete framing of assumptions or drop. Ln 46 – Concern about impacts on methane emissions is the reason it is not necessarily limited**
- **Figure 1 is useful but not intuitive, needs redrawing so easily followed without looking at the caption.**
- **L81 – definition of CLR not clear – what does it mean “ in terms of CH₄ mitigation”**

- **L97 – assumes all H2 emissions from energy systems is leakage, yet there is venting also occurring in hydrogen value chains, e.g. electrolysis**
- **L 119 – Norway leak rates, better to reference empirical studies, there is ample evidence that inventory-based estimates are low (IEA largely relies on inventory data). For Norway use Foulds et al . 2022 ACP. Still shows low emission rates but believe higher than IEA – the paper references emissions being higher than inventory.**
- **L116 – same as Norway better to use more up to date empirical data. US national survey using TROPOMI data by Varon et al (Daniel Jacob’s group at Harvard) – either recently published or about to be published.**
- **Figure 2 – central to the paper. Hard to unpack, needs to be more intuitive. Probably two rather one figure.**
- **Ln 214-222 provides a good framing of the paper – it would be better placed at the beginning of the intro rather at the beginning of the discussion.**

Reviewer #1 (Remarks to the Author):

The manuscript titled 'Tropospheric Methane Enhancement by Hydrogen Fugitive Emissions' presents the results of modelling the effect of H₂ leakages on methane enhancement. The work estimates a critical leakage rate of H₂, the point at which displacement over fossil fuels bears no climate benefit.

The work follows on from recent works which estimate the climate impacts of H₂ emissions and provides an interesting perspective on the H₂ and methane budgets. While the work is interesting and topical given the high interest in both methane and H₂, the paper is not suitable for publication in Nature in its current state. The underlying leakage assumptions and data are not rigorous which is the main weakness of the paper. I outline this concern, and other in more detail below.

We thank the Referee for his/her review work and comments, which helped increase the quality of the manuscript in terms of both rigor and presentation of the results. Specifically, we are now providing a detailed rationale for the hydrogen and methane emission rates used in the study. This has also led to a slight modification of the blue H₂ scenario, which had only a minor impact on the quantitative results. Line numbers refer to the article without track changes.

- **H₂ leakage rate: it is unclear whether the authors only consider leaks of H₂ i.e. emissions from leaky equipment/processes, so excluding other emissions such as venting and other fugitives (irregular/malfunctioning equipment).**

We thank the Reviewer for raising this point (which was also raised by Reviewer 3). We clarified in the text that we account for all hydrogen emissions in the hydrogen supply chain. This is first stated in the introduction section (lines 37-40)

The H₂ molecule is minuscule and difficult to contain, so it is still largely unknown how much H₂ will leak in future supply chains. H₂ emissions will also occur due to venting, purging, and incomplete combustion (van Ruijven et al., 2011; Frazer-Nash Consultancy, 2022; Cooper et al., 2022).

To further clarify that we include all H₂ emissions, we now use the expression hydrogen emission intensity (HEI) instead of the expression hydrogen leakage rate that we used before. The hydrogen emission intensity is defined in the emission section (lines 111-113)

... and the hydrogen emission intensity (HEI), namely the percentage of H₂ produced that is lost to the atmosphere. Losses can occur due to venting, purging, incomplete combustion and leaks across the hydrogen supply chain.

Finally, also the title of the manuscript has been changed from *Tropospheric Methane Enhancement by Hydrogen Fugitive Emissions* to *Potential for Atmospheric Methane Enhancement with Hydrogen Economy*. In this way it should be clearer that we include all potential emissions of the hydrogen economy.

- **H₂ leakage rate: there is a very high level of uncertainty in what the H₂ emissions could be in potential future H₂ supply chains. It is unclear in the manuscript if the authors only consider H₂ emissions from production or consider entire supply chains. The authors use literature values to derive a range of 1-10% but given the uncertainty, and that it is not specified what parts or processes of the H₂ supply chain this covers, this is not appropriate and the authors should have attempted to model their own H₂ leakage rates. There are also more recent studies which estimate H₂ emissions from H₂ supply chains and values from these could have been used or their methods applied:**

<https://www.sciencedirect.com/science/article/pii/S004896972201717X> or

<https://www.gov.uk/government/publications/fugitive-hydrogen-emissions-in-a-future-hydrogen-economy>

We agree with the Reviewer that there is a large uncertainty about future H₂ emission in a H₂ economy. This is pointed out several times along the paper (lines 38-39, 116, 325-339). As explained in the response above, in the revised manuscript we have clarified that we account for all H₂ emissions occurring across the supply chain. We have also added to manuscript the references suggested by the Reviewer and improved the paragraph that introduces the assumptions for the average hydrogen emission intensity. The paragraph now reads (lines 114-122)

The HEI of the future global H₂ supply chain is very uncertain. Literature values range from 1% to 12% (Schultz et al., 2003, van Ruijven et al., 2011, Bond et al., 2011), but the upper bound is unlikely to occur at large scales because it would be both unsafe and too expensive. Recent empirical estimates for specific H₂ infrastructures suggest HEI's ranging from 0.1 to 6.9%, critically depending on the pathway of hydrogen production and transport (Cooper et al., 2022). To account for these uncertainties and to explore a broad spectrum of possible scenarios, here we vary HEI from 0 to 10% of total H₂ produced (Fig.2a).

In the revised manuscript we now also acknowledge the importance of disaggregating hydrogen emissions across the various segments of the supply chain (e.g., for safety reason, to address mitigation opportunities etc.). However, it is also important to notice that this does not apply to our work, where we need to estimate the final flux of H₂ to the atmosphere. In this perspective, the most useful and probably simplest solution is to use a range of possible values for an average hydrogen emission intensity of the global supply chain, as we did in our analysis.

- **Methane leakage rate: it is unclear which sections of the gas supply chain are included in the emission rates assumed. For blue H₂, the stages of production, gathering and boosting, processing and transmission would be what would typically be considered but this varies depending on how far away the SMR plant is from the gas field, the size of the gas field and the quality of the gas produced in the gas field.**

- **Methane leakage rate: the methane leakage rates assumed (0.2 to 3%, average of 1.5%) is unclear if the authors are considering the US only or a global scale. It is also unclear if only methane leakages are considered and if other methane fugitives, venting and incomplete combustions emissions are excluded. The methane emission rate varies significantly from country to country so if estimating on a global context, it is not appropriate to use values specific to a handful of countries.**

We have specified in the manuscript that we consider all methane emissions associated with blue hydrogen production. From text at lines 136-139:

For blue H₂, which is derived from methane reforming, the variation in the CH₄ sources not only accounts for the reduced consumption of fossil fuels but also for the methane emissions (venting, incomplete combustion, fugitive) associated with blue hydrogen production.

In the revised manuscript, we are now using overall CH₄ leakage rate of 2, 1 and 0.2%. The choice of these values is explained in the paragraph reported below, where we also acknowledge the large heterogeneity of leak rates among different countries and different operators within the same country. Please note that, like hydrogen emissions, while disaggregating methane emissions across the various segments of the supply chain may be important for several other purposes, only the globally integrated CH₄ emissions associated with H₂ production are needed for our analysis.

The precise average leak rate of the global natural gas supply chain remains uncertain. One of the reasons is that national inventories generally underestimate real emissions (Zhang et al., 2020; Alvarez et al., 2018; Shen et al., 2022; MacKay et al., 2021). More detailed studies relying on field measurements in the United States and Canada estimate average leak rates around 2% (Alvarez et al., 2018; MacKay et al., 2021; Shen et al., 2022), with large spatial heterogeneity between different operators (Lauvaux et al., 2022). Although national inventories suggest that some countries, like Venezuela and Turkmenistan, have higher leak rates (IEA, 2022), here we adopt 2% as maximum global CH₄ leak rate for our scenarios, because methane-mitigation efforts are likely to decrease future global leak rates (European Commission, 2021) and, more importantly, because not all hydrogen produced will be blue H₂. In this regard, the scenario of blue H₂ with 2% CH₄ loss rate can also be interpreted as a combination of equal production of green H₂ and blue H₂ with 4% CH₄ loss rate. We use 0.2% as a lower bound for the CH₄ leak rate, since this has been declared as the target of several energy companies for 2025 (UNEP, 2021). 1% represents an intermediate scenario of blue H₂ production.

Also, it is confusing whether coal is considered as blue H₂ is typically natural gas SMR with CCS- referring to page 4 lines 107 to 122.

We have deleted those lines to avoid confusion.

• Methane leakage rate: the 3% assumed is from the paper by Howarth and Jacobson in their blue H₂ paper. It is unclear in the Howarth and Jacobson paper where this 3% comes from and what it is representative of. In the paper they cite the 3.5% emissions to be based on top-down estimates for emissions from 20 different studies in 10 different gas fields plus a top-down estimate for emissions from gas transport and storage. However, when the citation for this value is checked the reference given is for a book which is not yet published so it is uncertain what the 3/3.5% is representative of.

We agree with the Reviewer that the reference reported by Howarth and Jacobson is incomplete. We have deleted that reference and improved the scenarios of blue H₂ production as explained above.

• H₂ production: the projection by the Hydrogen Council was used but there are other studies which estimate future H₂ demand, such as the IEA and IPCC. Why were these not used and a range in H₂ production not considered?

Following the Referee's suggestion, we have included the range of estimates of rise in H₂ production in net-zero scenarios by the Hydrogen Council, the International Energy Agency (IEA), and the International Renewable Energy Agency (IRENA). These estimates are all relatively close, with H₂ production expected to rise from current 90 Tg/year to 530-660 Tg/year in 2050 (text at lines 161-165). We have considered an increase of 500 Tg/year in the global H₂ demand for the results of Fig. 3 (previously Fig. 2c-d). Note that these results are fairly insensitive to this choice since the atmospheric response to different production pathways scales almost linearly with the amount of H₂ produced (Fig. 2 and Extended Data Fig. 6).

• The paper would benefit from placing their results in context of impact on meeting climate goals. This would broaden the applicability of the results of the work and who would benefit from the work. In the Methods, a H₂ GWP₁₀₀ was estimated, which could have been applied to the estimated H₂ emissions. Also, given that H₂ perpetuates the impacts of methane, could a GWP for methane not have been calculated which takes into account the effect of a prolonged atmospheric lifespan?

We thank the Referee for raising this point. To address it, we have added in the discussion sections (lines 298-315) a paragraph where we evaluate the radiative forcing impact of hydrogen emissions to understand how much of the climate benefits of fossil-fuel displacement could be offset by the hydrogen economy. The results show that hydrogen emissions could have a radiative forcing impact over a 100-year time horizon of 1 to 12% of the CO₂ emissions from fossil fuels. Methane emissions for blue H₂ production could account, in the worst scenario, for another 10% offset of the climate benefits of the hydrogen economy. We have then expanded the discussion to remark the importance of minimizing gas losses to maximize the climate benefit of hydrogen use (lines 316-324).

We used the GWP100 by Warwick et al., 2022 for hydrogen, since this has been obtained with a more detailed atmospheric chemistry model that also accounts for hydrogen effects on ozone and stratospheric water vapor. Our estimate for the GWP of H₂ only accounts for the methane feedback. We stress that GWP of a gas is defined for the atmospheric consequences, in terms of radiative forcing imbalance, of a gas emission pulse. Hence the (indirect) GWP of H₂ accounts for the prolonged atmospheric lifetime of methane that follows an H₂ pulse.

• The authors considered a pulse of H₂ emissions. In the recent paper by Okko and Hamburg, they argue that this is not appropriate for H₂ given the likely emissions profile of emissions from H₂ supply chains. They argue continuous emissions rather than a pulse would better reflect the climate impacts of H₂.

We consider a pulse of H₂ only in the section of the Methods *Linear stability and transient dynamics*. This shows the transient effect of methane due to any H₂ emission to the atmosphere. In the main text we instead provide equilibrium scenarios that follow fixed and continuous emissions. We have further emphasized this in the revised version of the manuscript to make sure that this point is evident (lines 88-93).

• The paper has grammatical mistakes and typos throughout, and sections are difficult to read. Please have the paper proofread by a native English speaker.

Thanks for the comment. We carefully edited and proofread the revised version to avoid these mistakes.

• Figure 1: could an average value be added to the flux estimates as some of the min-max ranges are large?

Following this suggestion and the comment by Referee 3, we have redrawn Fig. 1, making it clearer and more intuitive. In the new version of the figure, best estimates of current fluxes are also reported.

• Page 4 line 93-94: 'or increase (Δ SCH₄>0), due to an additional methane demand for blue H₂ production' this is unlikely given initiatives to cut methane emissions through efforts to improve emissions detection e.g. OGMP2.0, EU methane regulation. Also, if traditional demands for natural gas dropped but demand for blue H₂ increased, would this result in Δ SCH₄>0?

We have deleted the sentence and made the paragraph clearer. We also better acknowledge in the text that there are global initiatives to reduce methane leak rates (lines 318-320). However, we do not think that this will necessarily lead to a global reduction in methane emissions. Several energy scenarios foresee an increase in the global use of natural gas. As a result, even if this increased gas demand comes with reduced leakage rates, there might still be an overall rise in methane emissions.

Regarding the second question, to deliver the same amount of energy with methane or with blue H₂ obtained from methane, more methane is needed in the second case (due to energy and feedstock losses in the SMR process). Hence the substitution of methane with blue H₂ could cause an increase in methane emissions.

Reviewer #2 (Remarks to the Author):

Review of ‘Tropospheric Methane Enhancement by Hydrogen Fugitive Emissions’ by Bertagni et al.

This paper uses a simple model to investigate the response of atmospheric methane to H₂ leakage from a potential future hydrogen energy industry. The impact of H₂ leakage on stratospheric water vapor and tropospheric ozone, which has been estimated by more complex models with detailed chemistry to represent about 50% of the indirect radiative forcing from hydrogen, is excluded from the analysis. Despite focusing only on the impact of H₂ on methane, the use of the simple model presented here has the advantage that it is possible to explore more of the parameter space within the significant certainties associated with future leakage rates and the present-day budget of atmospheric hydrogen.

This study describes a method for deriving a critical leakage rate of H₂ in terms of the impact on methane abundance, taking into account the reduction in methane emissions through displacement of fossil fuels by hydrogen and the chemical feedback of H₂ leakage on the methane lifetime. The results are timely and policy-relevant given the current interest in more widespread use of H₂ as part of commitments to reach net zero.

I believe the methodology is sound, and the paper is well structured and written and I would recommend for publication after the following minor comments are addressed.

We thank the Referee for his/her positive assessment of the manuscript and the thoughtful review, which helped us refine some evaluations and improve the quality of the manuscript. Please find below the point-to-point response. Line numbers refer to the manuscript without track changes.

I think it could be worth highlighting in the discussion that additional impacts of H₂ leakage on O₃ and H₂O could have further negative effects on the atmosphere, in terms of both air quality and radiative forcing, that are not considered. Also not included, is the impact of displacement of the fossil fuel industry on CO emissions. With reduced fossil fuel usage, CO emissions are also likely to reduce, which will feedback on the CH₄ abundance. Perhaps also a mention that any changes in nitrogen oxide emissions (either increases or decreases) will also influence OH, and thus the ultimate methane abundance in a future hydrogen economy.

Following the Referee’s suggestion, we have added in the discussion that H₂ emissions will also impact ozone and stratospheric water vapor, with implications for air quality and global warming (lines 299-302). The sentence introduces the new paragraph where we quantify and discuss the overall radiative forcing impact of the hydrogen economy.

At the end of the discussion section, we have also added a comment about the importance of addressing other emission changes, like NO_x and CO, for a more complete assessment of future CH₄-H₂-OH dynamics (lines 340-346).

Line 18: typo, should read ‘not only IS the CLR much lower’

The abstract has been rephrased and the previous typo fixed. Thank you.

Line 28: If H₂ is combusted, NO_x emissions may increase rather than be reduced relative to the combustion of fossil fuel (e.g. Lewis et al. 2021).

The first paragraph of the introduction has been rephrased and the previous sentence deleted. We have added a comment about NO_x emission change at the end of the discussion section.

Lines 96-103: In the present day H₂ budget, the fossil fuel industry is estimated to account for approx. 25% of the total H₂ source. It's not completely clear whether the change in H₂ emissions estimated in this study includes a reduction in H₂ emissions from the fossil fuel industry displaced by H₂, as well as the additional H₂ from leakage in an H₂ energy industry – or if it is only the additional H₂ leakage considered. Could the authors clarify?

In the previous version of the manuscript, we did not account for the reduction in H₂ emissions from the fossil-fuel displacement. The reason being that direct H₂ emissions from fossil fuels are small (11 - 20 Tg per year) compared to the potential H₂ emissions in a hydrogen economy (20 - 360 Tg per year). Nonetheless, we agree with the reviewer that accounting for this emission reduction is more correct and we have hence included it in our evaluations. This has been made clearer in the newly improved Fig. 1 and in the text at lines 108-109.

This fossil-fuel displacement reduces both CH₄ and H₂ emissions (Fig.1)

and at lines 124-125

With a perfectly sealed hydrogen supply chain, H₂ emissions would only decrease due to the lower fossil-fuel use (Fig. 2a).

Accordingly, also the analytical expression for the critical hydrogen emission intensity (eq. 15 in the manuscript) now accounts for the reduction in hydrogen emissions from the fossil fuel displacement. We point out that this newly added reduction accounts for the H₂ emissions from fossil fuels through the water-gas shift reaction, which are estimated to be around 15% of total H₂ source (Ehhalt et al., 2009. Paulot et al., 2021). The Reviewer refers to 25% probably accounting also for the production of atmospheric H₂ that follows fossil-fuel emissions of methane. This indirect effect was already accounted for through the reduction in the fossil-fuel methane emissions and the coupled H₂-CH₄ atmospheric dynamics.

Lines 108-109: Current global CH₄ emissions from the fossil fuel industry are uncertain (e.g. Schwietzke et al. 2016) – could be worth pointing out this will be another source of uncertainty in the derived CLR.

The Referee is right. We have added a paragraph explaining how the uncertainty in the fossil-fuel methane emissions affects the evaluation of the critical H₂ emission intensity (lines 262-265). The reference suggested by the Referee has also been added to the manuscript.

Lines 111-122: Could the authors clarify whether the energy efficiency of the CH₄ to H₂ SMR process is taken into account when determining the amount of CH₄ required to generate the H₂ in the 'blue' scenarios? And if so, what energy efficiency is assumed?

We thank the Referee for raising this point, which has led to a revision of the amount of methane needed for blue hydrogen production. In the previous version of the manuscript, we only accounted for the methane to be used as feedstock based on stoichiometry (2 kg of CH₄ for kg of H₂). We now account for

the methane to be used both as feedstock and as energy input for the SMR process (3.2 kg of CH₄ per kg of H₂). This has been made clear in the text at lines 436-439.

To obtain the value of r [amount of CH₄ needed per unit of H₂], we used the estimate by Collodi et al., (2017) of 3.7 kg of natural gas for kg of H₂, which includes feedstock and energy requirements, and we assumed that 85% of natural gas by weight is composed by methane.

Lines 164-172: Presumably these increases in equivalent CO₂ are still much smaller than the CO₂ emission reductions expected from the reduction in size of the fossil fuel industry – is it possible to give some idea of how these numbers compare?

We have added a paragraph in the discussion section (lines 299-315) where we compare the radiative forcing impact of the hydrogen economy with that of the CO₂ emissions from fossil fuels. The results show that hydrogen emissions could have a radiative forcing impact over a 100-year time horizon of 1 to 12% of the CO₂ emissions from fossil fuels. Methane emissions for blue H₂ production could account, in the worst scenario, for another 10% offset of the climate benefits of the hydrogen economy. We also point out in the same paragraph how, since H₂ and CH₄ are short-lived gas compared to CO₂, the values for the radiative forcing impact for a 100-year time horizon hide a much greater potential of global warming in the short term. We have then expanded the discussion to highlight the importance of minimizing gas losses to maximize the climate benefit of hydrogen use (lines 316-324).

Reviewer #3 (Remarks to the Author):

Bertagni et al provide new insights into the climate implications of shifting from a fossil fuel economy to a hydrogen economy. Their focus on the net impacts on tropospheric methane is innovative and useful. They provide new analyses which help to explore critical tradeoffs that have received limited attention within the literature. The approach is solid, but it is relatively hard work for the reader to get access to the bigger picture characterizing this work and how it fits into the larger issues associated with decarbonization strategies and how effective they are likely to be. The work being reported addresses several of the issues that need to be considered, but not all of them. That is not inherently a problem provided the authors more clearly articulate up front where their work fits into the larger set of issues that need to be considered. The paper would benefit from a broader initial framing and a clearer explanation of the role of the analysis they conduct on informing the decisions that will need to be made and why.

Overall, the paper is well written and would expect it could be of interest to a wide audience if written more accessibly. It is a topic that is of interest to many outside of the scientific community. As currently structured the paper would be an unnecessarily hard read for a non-scientific audience. In order to make it more accessible it would be useful to include some text describing how the CLR is policy relevant and can and cannot be used to understand under what conditions it might make sense to proceed with replacing fossil fuels with hydrogen and the criteria that might be deployed. Such a discussion would help the reader better understand what the CLR could be used for and for what type of questions it would not be relevant. These issues are left unaddressed which will result in confusion on the part of the many, if not most readers.

The paper implicitly assumes that emissions of hydrogen and methane are static, yet there is every reason to believe they are easily malleable given the emerging data on methane emissions, where differences among operators can be huge within the same production basin. I am sure the authors are aware of this, but readers may not. Given that the regulatory requirements/performance standards for methane emissions from natural gas value chains and hydrogen production are currently being debated in many parts of the world, highlighting the need and opportunity to minimize these emissions to capture the decarbonization potential of hydrogen seems important.

We thank the Referee for his/her review and comments, which helped improve the presentation of the manuscript. Specifically, we have worked to simplify the text and figures to make the manuscript more accessible to a non-scientific audience. We have also restructured introduction and discussion sections to include: a broader initial framing about hydrogen potential for decarbonization (lines 22-34 and 267-276); quantitative considerations on the offset of the climate benefits of the hydrogen economy by hydrogen and methane emissions (lines 299-315); a paragraph that discusses the need and opportunities of minimizing both methane and hydrogen emissions to maximize the climate benefits (lines 316-339). Line numbers refer to the article without track changes.

We hope that the newly simplified text and the additions of these sections may help the reader to get an intuitive understanding of our work and the implications of future hydrogen production for the energy transition. We thank again the Referee for pointing out this criticality.

Specific comments:

- **The title could be more direct, which would help to orient the reader a bit better i.e. Potential for Tropospheric CH₄ Enhancement with a Hydrogen Economy.**

We like the title suggestion of the Referee. We have accordingly changed the title of our manuscript to *Potential for Atmospheric Methane Enhancement with a Hydrogen Economy*

- **Abstract – description of CLR not clear, clear after reading the paper but not as written in the abstract**

We have rephrased the whole abstract to make it clearer and more accessible.

- **Ln 45 – statement that “This indirect radiative forcing is expected to be limited” is too general and not useful. Seems like such a statement is highly subjective and scenario dependent. A more complete framing of assumptions or drop. Ln 46 – Concern about impacts on methane emissions is the reason it is not necessarily limited**

The paragraph has been restructured and the sentence deleted. Furthermore, there is new paragraph in the discussion section where we quantify the radiative forcing impact of the hydrogen economy and discuss the implications for the energy transition. We find that hydrogen emissions could have a radiative forcing impact over a 100-year time horizon of 1 to 12% of the CO₂ emissions from fossil fuels. Methane emissions for blue H₂ production could account, in the worst scenario, for another 10% offset of the climate benefits of the hydrogen economy. We have then expanded the discussion to remark the importance of minimizing gas losses to maximize the climate benefit of hydrogen use (lines 316-339).

- **Figure 1 is useful but not intuitive, needs redrawing so easily followed without looking at the caption.**

We thank the Referee for pointing this out. The figure has been redrawn and should now appear more intuitive, without the need of reading the caption.

- **L81 – definition of CLR not clear – what does it mean “ in terms of CH₄ mitigation”**

We have restructured the paragraph, which should now appear clearer. In addition, we have changed the acronym definition from CLR (critical leakage rate) to critical HEI (hydrogen emission intensity). This way it should be clearer that we consider all H₂ emissions across the supply chain and not only H₂ leakages.

- **L97 – assumes all H₂ emissions from energy systems is leakage, yet there is venting also occurring in hydrogen value chains, e.g. electrolysis**

The Referee is right, and the same comment has also been raised by Referee 1. We have better defined in the text that our hydrogen emissions include intended actions, like venting and purging, as well as unintended losses due to incomplete combustion and leaks. This has been made clearer in the text several times, starting from the introduction at lines 35-48

The H₂ molecule is minuscule and difficult to contain, so it is still largely unknown how much H₂ will leak in future supply chains. H₂ emissions will also occur due to venting, purging, and incomplete combustion (van Ruijven et al., 2011; Frazer-Nash Consultancy, 2022; Cooper et al., 2022).

- **L 119 – Norway leak rates, better to reference empirical studies, there is ample evidence that inventory-based estimates are low (IEA largely relies on inventory data). For Norway use Foulds et al. 2022 ACP. Still shows low emission rates but believe higher than IEA – the paper references emissions being higher than inventory.**
- **L116 – same as Norway better to use more up to date empirical data. US national survey using TROPOMI data by Varon et al (Daniel Jacob's group at Harvard) – either recently published or about to be published.**

Following this comment and the observations of Referee 1, we have better justified our assumptions for the methane leak rates associated with blue H₂ production. Specifically, we have made clearer that we use values that should be representative of global averages and not country averages. We have also added a sentence about the general underestimation of methane emissions by national inventories, as correctly pointed out by the reviewer. The paragraph now reads (lines 137-154)

The precise average leak rate of the global natural gas supply chain remains unclear. One of the reasons is that national inventories generally underestimate real emissions (Zhang et al., 2020; Alvarez et al., 2018; Shen et al., 2022; MacKay et al., 2021). More detailed studies relying on field measurements in the United States and Canada estimate average leak rates around 2% (Alvarez et al., 2018; MacKay et al., 2021; Shen et al., 2022), with large spatial heterogeneity between different operators (Lauvaux et al., 2022). Although national inventories suggest that some countries, like Venezuela and Turkmenistan, have higher leak rates (IEA, 2022), here we adopt 2% as maximum global CH₄ leak rate for our scenarios. The reasons being that methane-mitigation efforts are likely to decrease future global leak rates (European Commission, 2021) and, more importantly, that not all hydrogen produced will be blue H₂. In this regard, the scenario of blue H₂ with 2% CH₄ loss rate can also be interpreted as a combination of equal production of green H₂ and blue H₂ with 4% CH₄ loss rate. We use 0.2% as a lower bound for the CH₄ leak rate, since this has been declared as the target of several energy companies for 2025 (UNEP, 2021). 1% represents an intermediate scenario of blue H₂ production.

- **Figure 2 – central to the paper. Hard to unpack, needs to be more intuitive. Probably two rather one figure.**

We thank the Referee for pointing this out. The figure has been split into two figures to make the content more accessible to the readers. Furthermore, the new figure 2 has been redrawn and should now appear more intuitive.

- **Ln 214-222 provides a good framing of the paper – it would be better placed at the beginning of the intro rather at the beginning of the discussion.**

This is a good suggestion; we agree. Accordingly, we have rephrased this paragraph as well as the beginning of the discussion and introduction sections.

Reviewer #1 (Remarks to the Author):

I am happy with the revisions made to the paper and the replies to the Reviewers' comments. Overall, I think the paper is now in a state it can be considered for publication in Nature Communications, but there are some minor revisions which need to be addressed before it can be published:

Title: the authors have changed the title to better reflect the work presented. However, the title does not make sense. I think 'The potential for atmospheric methane enhancement because of a hydrogen economy' or something similar would be clearer.

CH4 vs methane and H2 vs hydrogen. The authors switch between the two throughout the paper, e.g. line 68 'weakening CH4's removal and increasing CH4's' and then in line 69 'Second, methane is' and line 70 'a primary precursor of hydrogen' and then in line 71 'whose photolysis produces H2'.

Reviewer #2 (Remarks to the Author):

Review of 'Potential for Atmospheric Methane Enhancement with a Hydrogen Economy'

This version of the manuscript is much improved and the authors have addressed my initial comments. I would recommend for publication if my further comments below can be addressed.

Abstract: Given that there are some fairly significant uncertainties currently in both the present day H2 and CH4 budgets which will impact the results of this paper, as well as how exactly CH4 emissions may change etc., I would suggest including the uncertainties on the critical leakage rates in the abstract. More detailed studies are needed (e.g. 3D models with more comprehensive chemistry, better evaluation of potential changes in emission inventories).

Line 38: Is the H2 molecule 'minuscule', e.g. in comparison to CH4?

Lines 61-63: As methane increases (in addition to H2 increases) indirectly impact ozone and stratospheric water vapour, the climate impact of H2 emissions 'related to the perturbation of tropospheric methane', will include not only all of the impact of the CH4 perturbation, but also the part of the O3 and stratospheric water vapour perturbations that arise as a result of the methane perturbation. I.e. it will be greater than 50% (based on Paulot et al. 2021's numbers).

Lines 74-80: The authors discuss grey, blue and green hydrogen. Although blue and green H2 are the focus of this paper, there are now many colours referring to different methods of H2 production. If space allows, it might be useful to very briefly mention other options here (e.g. orange hydrogen, Osselin et al. 2022).

Lines 133-134: Could the authors provide a reference for the equal distribution of methane emissions among the coal, oil and gas sectors? Saunio et al. does not separate oil and gas, and given that the entire fossil fuel methane source is scaled to determine the methane emission reduction, this is an important assumption (H2 being used primarily to replace gas). This method to determine the methane emission reduction is quite basic which arguably can be justified by the fairly large uncertainties in the present day methane budget – I think it would be worth mentioning that a fuller assessment should include more detail here.

Lines 137-153: The calculated methane emissions resulting from H2 production depend upon more than just the assumed leakage rate (also 'r' discussed in Methods). A brief mention of this would be useful here, or at least to specify that more details on how

methane emission changes are determined is available in Methods etc..

Lines 202-214: 'The indirect H₂ radiative efficiency has been halved to not double count its feedback on CH₄ (Paulot et al. 2021)'. My comment here is related to my comment on lines 61-53. As the oxidation of both hydrogen and methane produces tropospheric O₃ and stratospheric water vapour, some of the indirect impact of H₂ on O₃ and H₂O arises as a result of the impact of H₂ on methane.

By halving the H₂ radiative efficiency from Paulot et al., only the impact of changes in the methane mixing ratio have been removed from the H₂ radiative efficiency – not the impact of those changes in methane mixing ratio (resulting from changes in H₂) on tropospheric O₃ and stratospheric H₂O.

Also, from the way this is written, it seems that the authors are accounting for the indirect impact of H₂ on tropospheric O₃ and stratospheric H₂O in their climate forcing calculations (by halving the H₂ radiative efficiency from Paulot et al.), but not the indirect impact of CH₄ on O₃ and H₂O resulting from any changes in CH₄ emissions (by using the Forster et al. methane radiative efficiency).

Both these points indicate some lack of consistency – could the authors clarify that all forcings have been correctly accounted for/which forcings are included? This paragraph is important as it attempts to quantify the overall pros/cons of switching to a hydrogen economy.

It might be useful to the reader if the authors could provide another example of how significant the determined ppm changes in CO₂ are, e.g. compare to the CO₂ change in ppm since the pre-industrial.

Line 420: The authors say that their overestimation in GWP is likely caused by their high CH₄ feedback factor of 1.5 – note the feedback factor in Warwick et al. was very similar, 1.49, and therefore this is unlikely to explain the difference.

Reviewer #3 (Remarks to the Author):

The revised manuscript by Bertagni et al. is much improved. It reads well, the figures are easy to understand as well as the messages more direct and impactful. My previous comments have been effectively addressed. There were two areas that are now more apparent with the improved text and where additional attention is still required:

- The use of the term supply chain is muddled with the term value chain. e.g. line 18 reference to supply chain should be value chain. Value chain includes the end use and supply chain does not. This is important as end use emissions of both hydrogen and methane can/could be very large. The authors need to go through the manuscript and sort out which is meant where. Reference to 2% loss rates for natural gas refers to supply chains but they need value chain numbers – given the end use for hydrogen production they are probably very similar – but US value chain natural gas emissions for urban uses are considerably higher than 2%. The paper requires much greater clarity with respect to these issues.

- The paper uses GWP₁₀₀ as the basis of the analysis. Yet, the authors imply that the results are policy relevant in ways that are spurious. They refer to net zero goals, e.g., line 272, yet those goals are almost exclusively focused on 2050, which requires looking at the 30-year time horizon, which will vary considerably from the 100 year results. The HEI as calculated is not compatible with most net zero considerations and that is not clear to the reader. If they want the HEI to be policy relevant, they need to address how HEI varies over time. Otherwise, they need to be explicit that the HEI will vary depending on the time horizon of interest and as presented the HEI only applies to answering questions of how the atmosphere and climate will be impacted over the next 100 years. It is important that the manuscript address this issue, or the HEI is likely to be misapplied to near term objectives such as to create misunderstandings.

Reviewer #1 (Remarks to the Author):

I am happy with the revisions made to the paper and the replies to the Reviewers' comments. Overall, I think the paper is now in a state it can be considered for publication in Nature Communications, but there are some minor revisions which need to be addressed before it can be published:

Title: the authors have changed the title to better reflect the work presented. However, the title does not make sense. I think 'The potential for atmospheric methane enhancement because of a hydrogen economy' or something similar would be clearer.

CH₄ vs methane and H₂ vs hydrogen. The authors switch between the two throughout the paper, e.g. line 68 'weakening CH₄'s removal and increasing CH₄'s' and then in line 69 'Second, methane is' and line 70 'a primary precursor of hydrogen' and then in line 71 'whose photolysis produces H₂'.

We are pleased by the Referee's positive judgement of the paper, and we thank him/her for the review work and comments.

Following the Referee's suggestion, we have changed the title to *Risk of The Hydrogen Economy to Enhance Atmospheric Methane*.

Regarding the use of the word (methane, hydrogen) vs the chemical formula (CH₄, H₂), we treat them as interchangeable, and we would prefer to maintain their combined use to avoid repetitiveness.

Reviewer #2 (Remarks to the Author):

Review of 'Potential for Atmospheric Methane Enhancement with a Hydrogen Economy'

This version of the manuscript is much improved and the authors have addressed my initial comments. I would recommend for publication if my further comments below can be addressed.

We are pleased that the Referee finds the work improved and we thank him/her for the constructive comments, which have helped revise some modeling results. Please find below the point-by-point response to the Referee's comments. Line numbers refer to the manuscript with track changes.

Abstract: Given that there are some fairly significant uncertainties currently in both the present day H₂ and CH₄ budgets which will impact the results of this paper, as well as how exactly CH₄ emissions may change etc., I would suggest including the uncertainties on the critical leakage rates in the abstract. More detailed studies are needed (e.g. 3D models with more comprehensive chemistry, better evaluation of potential changes in emission inventories).

We have included the estimated uncertainties of the critical HEI for green H₂ in the abstract. More broadly, we agree with the referee that more detailed studies are needed to refine our results. This is stated several times across the paper (e.g., final paragraph of the discussion). For the same reason, the section *critical HEI for methane mitigation* is devoted to the quantitative assessment of the possible source of uncertainties.

Line 38: Is the H₂ molecule 'minuscule', e.g. in comparison to CH₄?

We agree that the term *minuscule* may be an exaggeration, so we have substituted it with *very small* (line 39). Thank you.

Lines 61-63: As methane increases (in addition to H₂ increases) indirectly impact ozone and stratospheric water vapour, the climate impact of H₂ emissions 'related to the perturbation of tropospheric methane', will include not only all of the impact of the CH₄ perturbation, but also the part of the O₃ and stratospheric water vapour perturbations that arise as a result of the methane perturbation. I.e. it will be greater than 50% (based on Paulot et al. 2021's numbers).

The Referee is right that the impact on O₃ and stratospheric H₂O due to H₂ emissions is a superimposition of the H₂ and CH₄ effects. To our knowledge, however, there are not yet available literature data that precisely discriminate between the two components. To avoid confusion, we have deleted the to use 50% as per the reviewer's comment. The sentence (lines 65-66) now reads

Hence, H₂ emissions are far from being climate neutral, and their largest impact is related to the perturbation of CH₄ (Paulot 2021, Warwick 2022), the second most important anthropogenic GHG.

Lines 74-80: The authors discuss grey, blue and green hydrogen. Although blue and green H₂ are the focus of this paper, there are now many colours referring to different methods of H₂ production. If space allows, it might be useful to very briefly mention other options here (e.g. orange hydrogen, Osselin et al. 2022).

We have added references to other H₂ colors (white and orange) in the discussion section, clarifying that the results we obtained for green H₂ equally apply to other H₂ colors that do not involve fossil-fuel extraction. From lines 300-302

The same critical value would apply to other H2 colors that do not entail the use of fossil fuels, like white or orange H2 extracted from underground deposits (Prinzhofer 2018, Osselin 2022).

Lines 133-134: Could the authors provide a reference for the equal distribution of methane emissions among the coal, oil and gas sectors? Saunois et al. does not separate oil and gas, and given that the entire fossil fuel methane source is scaled to determine the methane emission reduction, this is an important assumption (H2 being used primarily to replace gas). This method to determine the methane emission reduction is quite basic which arguably can be justified by the fairly large uncertainties in the present day methane budget – I think it would be worth mentioning that a fuller assessment should include more detail here.

The IEA global methane tracker (link) estimates that methane emissions are almost equally distributed among coal, oil, and gas sectors. The reference has been added to the manuscript. We have also added at the end of the discussion section a sentence acknowledging that more refined analyses for the potential changes in emission inventories are need. From lines 367-368

Further analyses could also refine the potential changes in emission inventories due to H2 displacement of different energy sources.

Lines 137-153: The calculated methane emissions resulting from H2 production depend upon more than just the assumed leakage rate (also ‘r’ discussed in Methods). A brief mention of this would be useful here, or at least to specify that more details on how methane emission changes are determined is available in Methods etc..

We have added a sentence to clarify that the methane emissions depend on the amount of methane needed for H2 production in addition to the methane leakage rate. From lines 144-146:

These emissions depend on the amount of CH4 needed to produce H2, i.e., feedstock and energy requirements of the SMR process (Methods), and the CH4 leak rate.

Lines 202-214: ‘The indirect H2 radiative efficiency has been halved to not double count its feedback on CH4 (Paulot et al. 2021)’. My comment here is related to my comment on lines 61-53. As the oxidation of both hydrogen and methane produces tropospheric O3 and stratospheric water vapour, some of the indirect impact of H2 on O3 and H2O arises as a result of the impact of H2 on methane. By halving the H2 radiative efficiency from Paulot et al., only the impact of changes in the methane mixing ratio have removed from the H2 radiative efficiency – not the impact of those changes in methane mixing ratio (resulting from changes in H2) on tropospheric O3 and stratospheric H2O.

We agree with the Referee that there was a partial inconsistency in the CO2e results of Fig. 2. Specifically, the CO2e for CH4 accounted for both direct effect and indirect effects on O3 and stratospheric H2O. The CO2e for H2, with the radiative efficiency halved, accounted for indirect effects on O3 and H2O driven by both H2 and H2-induced CH4 perturbations. Namely, there was a partial double counting of CH4 indirect effects on O3 and H2O. We thank the Referee for pointing out this criticality.

We now present in Fig. 2 the CO2e only for the CH4 perturbation (both direct and indirect O3 and H2O effects). This is clearly stated at lines 214-215. Since the CH4 perturbation is already a result of the H2 emissions, the only climate impact left out is the direct effect of H2 on O3 and H2O. To the Authors’ knowledge, this cannot be precisely quantified as of now because the role of the CH4 vs H2 perturbation in the O3 and H2O impacts has not been clarified in the literature yet. Anyway, we consider this to not be a

problem since the climate impact of the direct effects of H₂ on O₃ and H₂O (<50% of H₂ impact) is expected to be very small when compared to the overall CH₄ impact (CH₄ having a radiative efficiency 3-4 times that of H₂).

Also, from the way this is written, it seems that the authors are accounting for the indirect impact of H₂ on tropospheric O₃ and stratospheric H₂O in their climate forcing calculations (by halving the H₂ radiatively efficiency from Paulot et al.), but not the indirect impact of CH₄ on O₃ and H₂O resulting from any changes in CH₄ emissions (by using the Forster et al. methane radiative efficiency).

As commented above, since we use the CH₄ radiative efficiency from Forster (2021) that accounts for direct and indirect effects, we were not missing the indirect effects but partially double counting them.

Both these points indicate some lack of consistency – could the authors clarify that all forcings have been correctly accounted for/which forcing are included? This paragraph is important as it attempts to quantify the overall pros/cons of switching to a hydrogen economy.

The paragraph has been clarified as explained above.

It might be useful to the reader if the authors could provide another example of how significant the determined ppm changes in CO₂ are, e.g. compare to the CO₂ change in ppm since the pre-industrial.

Following the Referee's suggestion, a sentence on the CO₂ change from pre-industrial time has been added to the paragraph, which now reads

For the same blue H₂, the rise in CH₄ following the entire displacement of fossil fuels would be like adding around 70 ppm of CO₂ (Fig. 2c). This is equivalent to around 50% of the CO₂ increase from preindustrial times (278 ppm) to current days (417 ppm).

Line 420: The authors say that their overestimation in GWP is likely caused by their high CH₄ feedback factor of 1.5 – note the feedback factor in Warwick et al. was very similar, 1.49, and therefore this is unlikely to explain the difference.

Our overestimation was due to the partial double counting explained above. We thank again the Referee for pointing out the issue, which has now been solved. The paragraph (lines 437-448) now reads

Using this result into traditional GWP formulas (Forster, 2021) yields a GWP for H₂ due to direct CH₄ perturbation around 7.8 with the 100-year time-horizon and 22 with the 20-year time horizon. It is estimated that around half of the H₂ indirect radiative forcing is due to the direct CH₄ perturbation, and the other half to the O₃ and stratospheric H₂O impacts caused by both H₂ and H₂-induced CH₄ perturbations (Paulot, 2021). Taking this into account yields a total GWP for H₂ of 15.6 with the 100-year time-horizon and 44 with the 20-year time-horizon. These values are in the upper range of the recent estimates of 11±5 for GWP₁₀₀ and 33⁺¹¹₋₁₃ for GWP₂₀ obtained with a detailed model of atmospheric chemistry (Warwick 2022).

Reviewer #3 (Remarks to the Author):

The revised manuscript by Bertagni et al. is much improved. It reads well, the figures are easy to understand as well as the messages more direct and impactful. My previous comments have been effectively addressed.

We are glad that the Referee finds the paper much improved, and we thank him/her for the constructive comments which have helped improving the presentation of the paper results. Line numbers refer to the manuscript without track changes.

There were two areas that are now more apparent with the improved text and where additional attention is still required:

- **The use of the term supply chain is muddled with the term value chain. e.g. line 18 reference to supply chain should be value chain. Value chain includes the end use and supply chain does not. This is important as end use emissions of both hydrogen and methane can/could be very large. The authors need to go through the manuscript and sort out which is meant where. Reference to 2% loss rates for natural gas refers to supply chains but they need value chain numbers – given the end use for hydrogen production they are probably very similar – but US value chain natural gas emissions for urban uses are considerably higher than 2%. The paper requires much greater clarity with respect to these issues.**

We better discriminate in the manuscript between value and supply chains. Regarding blue H₂, since natural gas will be used to produce hydrogen at industrial level, we believe that CH₄ loss rates of the value and supply will be almost identical – as also pointed out by the Referee. Urban and domestic uses for natural gas are not of interest for our work. On the contrary, hydrogen end uses are very important in our assessment and for this reason we now refer to the H₂ value chain.

- **The paper uses GWP100 as the basis of the analysis. Yet, the authors imply that the results are policy relevant in ways that are spurious. They refer to net zero goals, e.g., line 272, yet those goals are almost exclusively focused on 2050, which requires looking at the 30-year time horizon, which will vary considerably from the 100 year results. The HEI as calculated is not compatible with most net zero considerations and that is not clear to the reader. If they want the HEI to be policy relevant, they need to address how HEI varies over time. Otherwise, they need to be explicit that the HEI will vary depending on the time horizon of interest and as presented the HEI only applies to answering questions of how the atmosphere and climate will be impacted over the next 100 years. It is important that the manuscript address this issue, or the HEI is likely to be misapplied to near term objectives such as to create misunderstandings.**

We thank the Referee for this comment that gives us the opportunity to clarify a crucial aspect of the paper results. The results in the main text, which focus on the concentration responses of atmospheric H₂ and CH₄ (Fig. 2,3,4), are obtained for equilibrium conditions. For the H₂-CH₄-OH system, the timescales to equilibrium are dictated by the gas average lifetimes: around 2 years for H₂, 12 years for CH₄, and 1 second for OH (see Tab. 1). This has been clarified at lines 178-179. Namely, all these gases have relatively short lifetimes, and this is the reason why equilibrium conditions are meaningful. Our definition of critical HEI, i.e., the H₂ loss rate that offsets H₂ replacement of fossil fuels in methane mitigation, is also obtained for equilibrium conditions and hence does not depend on the time horizon.

By contrast, when considering the climate impacts of H₂ replacement of fossil fuels (lines 314-336 in the discussion section), timescales are important because CH₄ and H₂ are very short-lived compared to CO₂ (centuries). In this perspective, we agree with the Referee that the sole analysis with GWP100 was hard to interpret in terms of net-zero goals. We have hence also included an analysis with the GWP20 when

comparing the radiative forcing of hydrogen-based and fossil fuel-based energy systems (lines 314-336).
We hope that these results can be more policy relevant for net-zero goals.